# META-GUIDED DIFFUSION MODELS FOR ZERO-SHOT MEDICAL IMAGING INVERSE PROBLEMS

## ABSTRACT

In medical imaging, inverse problems aim to infer high-quality images from incomplete, noisy measurements, aiming to minimize expenses and risks to patients in clinical settings. The Diffusion Models have recently emerged as a promising approach to such practical challenges, proving particularly useful for the zero-shot inference of images from partially acquired measurements in Magnetic Resonance Imaging (MRI) and Computed Tomography (CT). A central challenge in this approach, however, is how to guide an unconditional prediction to conform to the measurement information. Existing methods rely on deficient projection or inefficient posterior score approximation guidance, which often leads to suboptimal results. In this paper, we propose a Meta-Guided Diffusion Model (MGDM) that tackles this challenge through a *bi-level* guidance strategy, where the *outer level* solves a proximal optimization problem to impose measurement consistency and the *inner level* approximates the measurement-conditioned posterior mean as the initial prediction. Furthermore, we introduce a refinement phase, termed the 'discrepancy gradient', designed to reduce the distance between the outputs of the aforementioned levels, thereby acting as an effective regularizer to further enhance data consistency in the recovered samples. Empirical results on publicly available medical datasets in MRI and CT highlight the superior performance of our proposed algorithm, faithfully reproducing high-fidelity medical images consistent with measurements, and notably mitigating the generation of hallucinatory images observed in state-of-the-art methods under similar conditions.

## 1 INTRODUCTION

Contemporary diagnostic medicine highly relies on advanced, non-invasive imaging techniques, notably Magnetic Resonance Imaging (MRI) and Computed Tomography (CT). Their unparalleled accuracy in capturing detailed anatomical measurements is of paramount importance for identifying internal abnormalities. In MRI, the Fourier transform of the spatial distribution of proton spins from the subject is acquired as measurements, which is commonly referred to as 'k-space' in medical imaging contexts. In the case of CT, raw measurements, also known as 'sinograms', are derived from X-ray projections obtained at various orientations around the patient. However, full k-space and sinogram acquisitions in MRI and CT often require prolonged scan durations and may pose health risks due to increased heat and radiation exposures (Lustig et al., 2007; Brenner & Hall, 2007). In light of these implications, there have been ongoing efforts toward reducing the number of measurements, exemplified by undersampled k-spaces in MRI and sparse-view sinograms in CT. While advantageous in accelerating medical imaging procedures, sparsification and undersampling introduce difficulties in reconstructing accurate and high-quality images (Donoho, 2006).

Medical image reconstruction can be mathematically characterized as solving an ill-posed linear inverse problem (Arridge, 1999; Bertero et al., 2021). The linear inverse problem is formulated as recovering an unknown target signal of interest $\mathbf{x} \in \mathcal{X} \subseteq \mathbb{C}^n$ from a noisy observed measurement $\mathbf{y} \in \mathcal{Y} \subseteq \mathbb{C}^m$, given by $\mathbf{y} = \mathcal{A}\mathbf{x} + \mathbf{n}$, where $\mathcal{A} \in \mathbb{C}^{m \times n}$ is a matrix that models a known linear measurement acquisition process (a.k.a. forward operator $\mathcal{A} : \mathbb{C}^m \to \mathbb{C}^n$), and $\mathbf{n} \in \mathbb{C}^{m \times 1}$ is an additive noise, simply treated here to follow the Gaussian distribution $\mathbf{n} \sim \mathcal{N}(\mathbf{0}, \sigma_{\mathbf{y}}^2 \mathbf{I})$. If the forward operator $\mathcal{A}$ is singular, e.g., when $m < n$, the problem is ill-posed, indicating that the solution might not exist, be unique, or depend continuously on the measurements (O'Sullivan, 1986). To mitigate the ill-posedness, it is essential to incorporate an additional assumption based on *prior* knowledge to

constrain the space of possible solutions. In this manner, the inverse problem then can be addressed by optimizing or sampling a function that integrates this prior or regularization term with a data consistency or likelihood term (Ongie et al., 2020). A prevalent approach for prior imposition is to employ pre-trained deep generative models (Bora et al., 2017; Jalal et al., 2021).

Diffusion Models (DMs) (Sohl-Dickstein et al., 2015; Song & Ermon, 2019; Ho et al., 2020; Song et al., 2020b) are a novel class of deep generative models (Yang et al., 2022) that have recently shown powerful capabilities in solving ill-posed inverse problems. These models are primarily designed to encode implicit prior probability distributions over data manifolds, represented as $\nabla_{\mathbf{x}} \log p(\mathbf{x})$. Once trained, they can be leveraged as a chain of denoisers to produce conditional samples at inference time in a zero-shot fashion (a.k.a. plug-and-play approach) (Zhang et al., 2021; Jalal et al., 2021; Chung et al., 2022a; Wang et al., 2022). This approach is particularly of significance in medical imaging, as measurement acquisitions can vary significantly upon such circumstances as instrumentations, scan protocols, acquisition time limit, and radiation dosage (Jalal et al., 2021; Song et al., 2021; Chung & Ye, 2022).

Top-performing methods that utilize DMs to tackle inverse problems in a zero-shot setting typically follow a three-phase progression in the iterative reverse diffusion process. Initially, they begin with an unconditional prediction, which might be either a transient noisy image (Song et al., 2021) or its denoised estimate version (Chung et al., 2022b;a; Song et al., 2022). The subsequent phase, crucial for conditional sampling, entails guiding the initial prediction with information drawn from observed measurements. This has been accomplished via projecting images into the measurement-consistent subspaces (Song et al., 2021; Lugmayr et al., 2022; Kawar et al., 2022; Wang et al., 2022), approximating posterior score towards higher time-dependent likelihood (Chung et al., 2022a; Meng & Kabashima, 2022; Feng et al., 2023; Fei et al., 2023; Mardani et al., 2023), and performing proximal optimization steps (Chung et al., 2023). While the radical projection might throw the sampling trajectory off the data manifold (Chung et al., 2022a), and subtle score approximation may fail to generalize well to fewer timesteps (Song et al., 2023), proximal optimization appears promising, particularly for medical imaging applications (Chung et al., 2023). Nonetheless, the efficiency of this iterative proximal gradient-based optimization significantly diminishes in the absence of a closed-form solution (Chung et al., 2023). Ultimately, in the third phase, the procedure progresses to the sampling, which is performed using Langevin dynamics (Song et al., 2020b; Ho et al., 2020) or more efficient samplers (Song et al., 2020a; Chung et al., 2022c).

In this paper, we introduce Meta-Guided Diffusion Models (MGDM), an approach that guides the diffusion process through a *bi-level* strategy, which leverages the unique strengths of different guidance mechanisms, aiming to provide a more effective and efficient way of measurement incorporation. To this end, we first theoretically examine the range-null space decomposition (Wang et al., 2022), a projection-based technique, from an optimization perspective, leading us to an alternative proximal optimization objective. This *outer-level* objective explicitly takes into account both data fidelity and proximity terms, where the former enforces that the reconstructed image is consistent with the acquired measurements in the transformed domains (k-space and sinograms), and the latter ensures that the solution remains close to its initial prediction estimated by the denoiser—the pre-trained DM. Notably, this optimization problem offers a closed-form solution. However, its effectiveness relies on a more accurate, and consistent initial prediction. To achieve this without deviation from the clean manifold, we propose to implement an *inner-level* estimate of the clean image conditioned on its noisy counterpart and the measurement. Furthermore, we introduce an additional phase named the 'discrepancy gradient', through which the generated samples from each reverse diffusion step are refined by gradient descent of the discrepancy between the bi-levels with respect to the transient noisy image. We empirically found that this adjustment further encourages data consistency, especially for the CT reconstruction task.

The contribution of our work is as follows. **In theory**, we delve into the effective strategies tailored for addressing medical imaging inverse problems in a zero-shot setting. At the core of our approach is an assurance of data consistency achieved through analytical measures complemented by the integration of prior information extracted from pre-trained diffusion models. **In practice**, our methodology is rigorously evaluated across a spectrum of challenges, including under-sampled MRI and sparse-view CT reconstructions. Empirical results consistently indicate that our approach surpasses the state-of-the-art performance benchmarks, exhibiting robustness across diverse acceleration rates, projection counts, and anatomical variations (human brains, lungs, and knees).

## 2    PRELIMINARIES

### 2.1    DIFFUSION MODELS

A diffusion model (Sohl-Dickstein et al., 2015) is composed of two processes with $T$ timesteps. The first is the *forward* noising process (*diffusion process*), which gradually introduces Gaussian noise into the data sample $\mathbf{x}_0 \sim q(\mathbf{x}_0)$. During this procedure, a series of latent variables $\mathbf{x}_1, ...\mathbf{x}_T$ are sequentially generated, with the final one, $\mathbf{x}_T$, roughly conforming to a standard Gaussian distribution, i.e., $q(\mathbf{x}_T) \approx \mathcal{N}(\mathbf{x}_T; \mathbf{0}, \mathbf{I})$. This process is formally defined as a Markov chain

$$q(\mathbf{x}_{1:T}|\mathbf{x}_0) = \prod_{t=1}^{T} q(\mathbf{x}_t|\mathbf{x}_{t-1}), \quad q(\mathbf{x}_t|\mathbf{x}_{t-1}) = \mathcal{N}(\mathbf{x}_t; \sqrt{1-\beta_t}\mathbf{x}_{t-1}, \beta_t \mathbf{I}), \tag{1}$$

where $q(\mathbf{x}_t|\mathbf{x}_{t-1})$ signifies the Gaussian transition kernel with a predefined variance schedule $\beta_t$. One can further compute the probabilistic distribution of $\mathbf{x}_t$ given $\mathbf{x}_0$ via reparametrization trick as $q(\mathbf{x}_t|\mathbf{x}_0) = \mathcal{N}(\mathbf{x}_t; \sqrt{\overline{\alpha}_t}\mathbf{x}_0, (1-\overline{\alpha}_t)\mathbf{I})$ with $\alpha_t = 1 - \beta_t$ and $\overline{\alpha}_t = \prod_{i=0}^{t} \alpha_i$. Equivalently, $\mathbf{x}_t$ can be expressed as $\mathbf{x}_t = \sqrt{\overline{\alpha}_t}\mathbf{x}_0 + \boldsymbol{\sigma}_t\boldsymbol{\epsilon}$, where $\boldsymbol{\sigma}_t = \sqrt{1-\overline{\alpha}_t}$ and $\boldsymbol{\epsilon} \sim \mathcal{N}(\mathbf{0}, \mathbf{I})$. The other is the *reverse* denoising process, which aims to recover the data-generating sample $\mathbf{x}_0$ by iteratively denoising the initial sample $\mathbf{x}_T$ drawn from standard Gaussian distribution $p(\mathbf{x}_T) = \mathcal{N}(\mathbf{x}_T; \mathbf{0}, \mathbf{I})$. This procedure is also characterized by the following Markov chain:

$$p_\theta(\mathbf{x}_{0:T}) = p(\mathbf{x}_T) \prod_{t=T}^{1} p_\theta(\mathbf{x}_{t-1}|\mathbf{x}_t), \quad p_\theta(\mathbf{x}_{t-1}|\mathbf{x}_t) = \int_{\mathbf{x}_0} q(\mathbf{x}_{t-1}|\mathbf{x}_t, \mathbf{x}_0)p_\theta(\mathbf{x}_0|\mathbf{x}_t)d\mathbf{x}_0, \tag{2}$$

where $p_\theta(\mathbf{x}_{t-1}|\mathbf{x}_t)$ is a denoising transition module with parameters $\theta$ approximating the forward posterior probability distribution $q(\mathbf{x}_{t-1}|\mathbf{x}_t) = q(\mathbf{x}_{t-1}|\mathbf{x}_t, \mathbf{x}_0)$. The objective is to maximize the likelihood of $p_\theta(\mathbf{x}_0) = \int p_\theta(\mathbf{x}_{0:T})d\mathbf{x}_{1:T}$. Denoising Diffusion Probabilistic Models (DDPM) (Ho et al., 2020) assumes $p_\theta(\mathbf{x}_{t-1}|\mathbf{x}_t) = \mathcal{N}(\mathbf{x}_{t-1}; \boldsymbol{\mu}_\theta(\mathbf{x}_t, t), \boldsymbol{\sigma}_\theta(\mathbf{x}_t, t)\mathbf{I})$ by considering $p_\theta(\mathbf{x}_0|\mathbf{x}_t)$ to be a Dirac delta distribution centered at the point estimate $\mathbb{E}[\mathbf{x}_0|\mathbf{x}_t]$, which is minimum mean squared error (MMSE) estimator of $\mathbf{x}_0$ given $\mathbf{x}_t$, and $q(\mathbf{x}_{t-1}|\mathbf{x}_t, \mathbf{x}_0)$ to be a fixed Gaussian. Under this scheme, the loss $\ell(\theta)$ can be simplified as

$$\min_\theta \ell(\theta) := \min_\theta \mathbb{E}_{t\sim(\mathbf{0},T),\mathbf{x}_0\sim q(\mathbf{x}_0),\boldsymbol{\epsilon}\sim\mathcal{N}(\mathbf{0},\mathbf{I})}\big[\|\boldsymbol{\epsilon} - \boldsymbol{\epsilon}_\theta(\mathbf{x}_t, t)\|_2^2\big]. \tag{3}$$

Therefore, given the trained denoising function $\boldsymbol{\epsilon}_\theta(\mathbf{x}_t, t)$, samples can be generated using DDPM, Denoising Diffusion Implicit Models (DDIM) (Song et al., 2020a), or other solvers (Lu et al., 2022; Zhang & Chen, 2022).

### 2.2    SOLVING LINEAR INVERSE PROBLEMS WITH DIFFUSION MODELS

An inverse problem seeks to estimate an unknown image $\mathbf{x}$ from partially observed, noisy measurement $\mathbf{y}$. They are generally approached by optimizing or sampling a function that combines a term for data fidelity or likelihood with a term for regularization or prior (Ongie et al., 2020). A detailed exploration of methods for solving linear inverse problems can be found in Appendix A.1. A common method for regularization involves using pre-trained priors from generative models. Recently, pre-trained diffusion models (Ho et al., 2020; Nichol & Dhariwal, 2021) have been leveraged as a powerful generative prior (a.k.a. denoiser), in a zero-shot fashion, to efficiently sample from the conditional posterior. Due to their unique characteristics, namely the ability to model complex, the efficient iterative nature of the denoising process, and the capacity to effectively conduct conditional sampling, these models stand out as a potent solution for solving inverse problems (Daras et al., 2022; Rombach et al., 2022). A primary difficulty, however, is how to guide the unconditional prediction to conform to the measurement information in each iteration. Methods addressing this generally fall into two distinct categories as follows.

**Posterior Score Approximation.** The reverse Stochastic Differential Equation (SDE) for a conditional generation can be written as

$$d\mathbf{x}_t = \big[\mathbf{f}(\mathbf{x}_t, t) - g^2(t)\nabla_{\mathbf{x}_t} \log p_t(\mathbf{x}_t|\mathbf{y})\big]d\bar{t} + g(t)d\bar{\mathbf{w}}_t, \tag{4}$$

where $\nabla_{\mathbf{x}_t} \log p_t(\mathbf{x}_t|\mathbf{y})$ is referred to as posterior score that can be decomposed through Bayesian' rule as follows.

$$\nabla_{\mathbf{x}_t} \log p(\mathbf{x}_t|\mathbf{y}) = \nabla_{\mathbf{x}_t} \log p(\mathbf{x}_t) + \nabla_{\mathbf{x}_t} \log p(\mathbf{y}|\mathbf{x}_t). \tag{5}$$

The composite score results from the prior score combined with the time-dependent likelihood score. While one can closely approximate the prior score with a pre-trained diffusion model, i.e., $\nabla_{\mathbf{x}_t} \log p(\mathbf{x}_t) \simeq \frac{-1}{\sqrt{1-\bar{\alpha}_t}} \boldsymbol{\epsilon}_\theta(\mathbf{x}_t, t)$, the likelihood score is analytically intractable to compute. This becomes evident when considering $p(\mathbf{y}|\mathbf{x}_t) = \int_{\mathbf{x}_0} p(\mathbf{y}|\mathbf{x}_0)p(\mathbf{x}_0|\mathbf{x}_t)\,d\mathbf{x}_0$ according to the graphical inferences $\mathbf{x}_0 \to \mathbf{y}$ and $\mathbf{x}_0 \to \mathbf{x}_t$. The measurement models can be represented by $p(\mathbf{y}|\mathbf{x}_0) := \mathcal{N}(\mathcal{A}\mathbf{x}_0, \boldsymbol{\sigma}_{\mathbf{y}}^2)$. The intractability of $p(\mathbf{y}|\mathbf{x}_t)$ arises from $p(\mathbf{x}_0|\mathbf{x}_t)$. Several strategies have been proposed to approximate the likelihood term. Among the most prevalent are DPS (Chung et al., 2022a) and ΠGDM (Song et al., 2022), where point-estimate $p(\mathbf{x}_0|\mathbf{x}_t) = \delta(\mathbf{x}_0 - \mathbf{x}_{0|t})$ and Gaussian assumption $p(\mathbf{x}_0|\mathbf{x}_t) \sim \mathcal{N}(\mathbf{x}_{0|t}, \sigma_t^2/\sigma_t^2{+}1\,\mathbf{I})$ are considered respectively to estimate $p(\mathbf{y}|\mathbf{x}_t)$. The term $\mathbf{x}_{0|t}$ is posterior mean (or denoised estimate) of $\mathbf{x}_0$ conditioned on $\mathbf{x}_t$, defined as $\mathbf{x}_{0|t} := \mathbb{E}[\mathbf{x}_0|\mathbf{x}_t] = \mathbb{E}_{\mathbf{x}_0 \sim p(\mathbf{x}_0|\mathbf{x}_t)}[\mathbf{x}_0]$. As a result, the likelihood score can be reformulated as

$$\nabla_{\mathbf{x}_t} \log p(\mathbf{y}|\mathbf{x}_t) \simeq \underbrace{\frac{\partial\,(\mathbf{x}_{0|t})}{\partial \mathbf{x}_t}}_{\text{J}} \underbrace{\mathcal{H}(\mathbf{y} - \mathcal{A}\,\mathbf{x}_{0|t})}_{\text{V}}, \tag{6}$$

which is essentially a Vector (V)-Jacobian (J) Product (VJP) that enforces consistency between the denoising result and the measurements, with $\mathcal{H}$ corresponding to $\mathcal{A}^\top$ in DPS and to $\mathcal{A}^\dagger$ (the Moore-Penrose pseudoinverse of $\mathcal{A}$) in ΠGDM. These methods efficiently handle inverse problems over extended timesteps, yet face challenges with shorter durations (Chung et al., 2023). Moreover, in the context of MRI reconstruction in medical imaging, DPS leads to noisy outputs (Chung et al., 2023). More recently, variational posterior approximation has been proposed (Mardani et al., 2023), yet it requires computationally expensive test-time optimization.

**Decomposition/Projection Based.** Denoising Diffusion Restoration Model (DDRM) (Kawar et al., 2022) attempted to solve inverse problems in a zero-shot way using singular value decomposition (SVD) of $\mathcal{A}$. However, for medical imaging applications with complex measurement operators, the SVD decomposition can be prohibitive (Chung et al., 2023). Song et al. (2021) proposed an alternative decomposition of $\mathcal{A}$ in the sampling process, suitable for medical imaging, assuming that $\mathcal{A}$ is of full rank. Denoising Diffusion Null-Space Models (DDNM) (Wang et al., 2022) introduces a range-null space decomposition for zero-shot image reconstruction, where the range space ensures data consistency, and the null space enhances realism. Both Song's method and DDNM essentially use back-projection tricks (Tirer & Giryes, 2020) to meet the measurement consistency in a non-noisy measurement scenario, which can be expressed as:

$$\hat{\mathbf{x}}_t = \sqrt{\bar{\alpha}_t}\big(\mathcal{A}^\dagger \mathbf{y} + (\mathbf{I} - \mathcal{A}^\dagger \mathcal{A})\mathbf{x}_{0|t}\big) + \boldsymbol{\sigma}_t \boldsymbol{\epsilon}, \tag{7}$$

where the extra noise $\boldsymbol{\sigma}_t \boldsymbol{\epsilon}$ is excluded in DDNM, yielding a higher performance. However, these projection-based methods frequently encounter challenges in maintaining the sample's realness, as the projection might shift the sample path away from the data manifold (Chung et al., 2022b).

## 3 METHOD

We motivate our approach by highlighting two critical drawbacks inherent in projection-based methods, especially in DDNM, which utilizes the range-null space decomposition to construct a general solution $\hat{\mathbf{x}}$ as

$$\hat{\mathbf{x}} = \mathcal{A}^\dagger \mathbf{y} + (\mathbf{I} - \mathcal{A}^\dagger \mathcal{A})\bar{\mathbf{x}}, \tag{8}$$

where $\bar{\mathbf{x}}$ can be chosen arbitrarily from $\mathbb{C}^n$ without affecting the consistency. The foundational interplay between these spaces is evident: the range space, represented by $\mathcal{A}^\dagger \mathbf{y}$, embodies the solution components originating from observations, whereas the null space, denoted by $(\mathbf{I} - \mathcal{A}^\dagger \mathcal{A})\bar{\mathbf{x}}$, encompasses the solution's unobserved elements. We illuminate a new interpretation of this decomposition from an optimization perspective in the following proposition, whose proof can be found in Appendix A.2.

**Proposition 3.1** *Consider the least squares problem* $\min_{\mathbf{x} \in \mathbb{R}^n} \|\mathbf{y} - \mathcal{A}\mathbf{x}\|_2^2$ *where* $\mathcal{A} \in \mathbb{R}^{m \times n}$ *is any matrix and* $\mathbf{y} \in \mathbb{R}^m$. *Gradient descent, initialized at* $\bar{\mathbf{x}} \in \mathbb{R}^n$ *and with small enough learning rate, converges to* $\hat{\mathbf{x}} = \mathcal{A}^\dagger \mathbf{y} + (\mathbf{I} - \mathcal{A}^\dagger \mathcal{A})\bar{\mathbf{x}}$.

**Algorithm 1** DDNM Sampling (Wang et al., 2022)

**Require:** The measurement $\mathbf{y}$, and the forward operator $\mathcal{A}$
1: $\mathbf{x}_T \sim \mathcal{N}(\mathbf{0}, \mathbf{I})$
2: **for** $t = T, \ldots, 1$ **do**
3:     $\overline{\alpha}_{t-1} \leftarrow 1 - \boldsymbol{\sigma}_t^2$
4:     $c_1 \leftarrow \eta\sqrt{1 - \overline{\alpha}_{t-1}}$
5:     $c_2 \leftarrow \sqrt{1 - \overline{\alpha}_{t-1} - c_1^2}$
6:     $\boldsymbol{\epsilon} \sim \mathcal{N}(\mathbf{0}, \mathbf{I})$ **if** $t > 0$, **else** $\boldsymbol{\epsilon} = 0$
7:     $\mathbf{x}_{0|t} \leftarrow \frac{1}{\sqrt{\overline{\alpha}_t}}\left(\mathbf{x}_t - \sqrt{1 - \overline{\alpha}_t}\boldsymbol{\epsilon}_\theta(\mathbf{x}_t, t)\right)$

8:     $\hat{\mathbf{x}}_{0|t} \leftarrow \mathcal{A}^\dagger \mathbf{y} + (\mathbf{I} - \mathcal{A}^\dagger\mathcal{A})\mathbf{x}_{0|t}$

9:     $\mathbf{x}_{t-1} \leftarrow \sqrt{\overline{\alpha}_{t-1}}\hat{\mathbf{x}}_{0|t} + (c_2\boldsymbol{\epsilon}_\theta(\mathbf{x}_t, t) + c_1\boldsymbol{\epsilon})$
10: **end for**

11: **return** $\mathbf{x}_0$

**Algorithm 2** MGDM Sampling

**Require:** The measurement $\mathbf{y}$, and the forward operator $\mathcal{A}$
1: $\mathbf{x}_T \sim \mathcal{N}(0, \mathbf{I})$
2: **for** $t = T, \ldots, 1$ **do**
3:     $\overline{\alpha}_{t-1} \leftarrow 1 - \boldsymbol{\sigma}_t^2$
4:     $c_1 \leftarrow \eta\sqrt{1 - \overline{\alpha}_{t-1}}$
5:     $c_2 \leftarrow \sqrt{1 - \overline{\alpha}_{t-1} - c_1^2}$
6:     $\boldsymbol{\epsilon} \sim \mathcal{N}(\mathbf{0}, \mathbf{I})$ **if** $t > 0$, **else** $\boldsymbol{\epsilon} = 0$
7:     $\mathbf{x}_{0|t} \leftarrow \frac{1}{\sqrt{\overline{\alpha}_t}}\left(\mathbf{x}_t - \sqrt{1 - \overline{\alpha}_t}\boldsymbol{\epsilon}_\theta(\mathbf{x}_t, t)\right)$
8:     $\tilde{\mathbf{x}}_{0|t} \leftarrow \mathbf{x}_{0|t} - \zeta\nabla_{\mathbf{x}_t}\|\mathbf{y} - \mathcal{A}\mathbf{x}_{0|t}\|_2^2$
9:     $\hat{\mathbf{x}}_{0|t} \leftarrow \arg\min_{\mathbf{x}} \frac{1}{2}\|\mathbf{y} - \mathcal{A}\mathbf{x}\|_2^2 + \frac{\lambda}{2}\|\mathbf{x} - \tilde{\mathbf{x}}_{0|t}\|_2^2$
10:     $\mathbf{x}_{t-1} \leftarrow \sqrt{\overline{\alpha}_{t-1}}\hat{\mathbf{x}}_{0|t} + (c_2\boldsymbol{\epsilon}_\theta(\mathbf{x}_t, t) + c_1\boldsymbol{\epsilon})$
11:     $\hat{\mathbf{x}}_{t-1} \leftarrow \mathbf{x}_{t-1} - \rho\nabla_{\mathbf{x}_t}\|\hat{\mathbf{x}}_{0|t} - \tilde{\mathbf{x}}_{0|t}\|_2^2$
12: **end for**
13: **return** $\mathbf{x}_0$

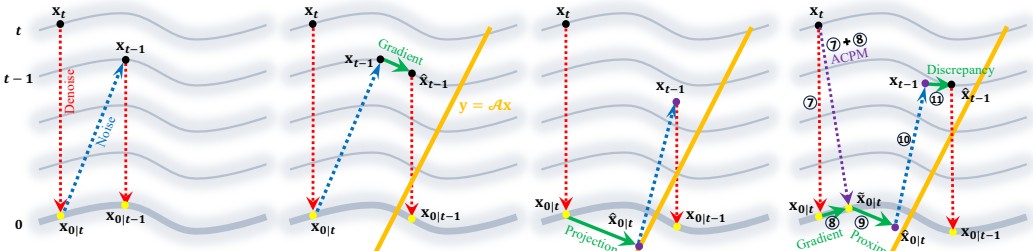

(a) DDIM (Song et al., 2020a) (b) DPS (Chung et al., 2022a) (c) DDNM (Wang et al., 2022)    (d) MGDM (**ours**)

Figure 1: An illustration of the geometric principles underpinning diffusion samplers and various guidance schemes. (a) DDIM is an unconditional diffusion sampler devoid of guidance. (b) DPS employs gradient guidance ensuring updated samples remain on the accurate manifold. (c) DDNM projects denoised samples into a measurement-consistent subspace. (d) Our proposed method employs a bi-level guidance strategy; the inner level approximates the initial prediction with a conditional posterior mean through gradient guidance, while the outer level tackles an optimization problem to further impose measurement consistency. Note that ACPM stands for Approximated Conditional Posterior Mean derived in Eq. (A.4.2).

Proposition 3.1 highlights the behavior of gradient descent on a least squares problem when initiated from any initial point, in particular $\bar{\mathbf{x}} = \mathbf{x}_{0|t}$. The solution, upon convergence, can be expressed as

$$\hat{\mathbf{x}}_{0|t} = \mathbf{x}_{0|t} + \mathcal{A}^\dagger(\mathbf{y} - \mathcal{A}\mathbf{x}_{0|t}). \tag{9}$$

Here, the term $\mathcal{A}^\dagger(\mathbf{y} - \mathcal{A}\mathbf{x}_{0|t})$ represents the correction applied to the initial estimate, factoring in the difference between predicted and observed measurements. However, this method is not devoid of challenges. The correction term, solely determined by $(\mathbf{y} - \mathcal{A}\mathbf{x}_{0|t})$, can be significantly affected if $\mathbf{y}$ is noisy, potentially leading our estimates astray. Furthermore, this correction direction, which is purely governed by the gradient of the discrepancy, can lead us to a suboptimal estimate, particularly when $\mathbf{x}_{0|t}$ itself holds uncertainties. To address these concerns, we define the decomposition Eq. (9) explicitly by embedding a regularization term into our optimization objective, acting as a penalty against large deviations from our initial estimate. This results in the following *outer-level* regularized objective:

$$\hat{\mathbf{x}}_{0|t} = \arg\min_{\mathbf{x}} \frac{1}{2}\underbrace{\|\mathbf{y} - \mathcal{A}\mathbf{x}\|_2^2}_{\text{Fidelity}} + \frac{\lambda}{2}\underbrace{\|\mathbf{x} - \mathbf{x}_{0|t}\|_2^2}_{\text{Proximity}}, \tag{10}$$

where the fidelity term aims to minimize the discrepancy between the predicted and observed measurements, while the proximity term penalizes deviations from the initial estimate. This is crucial, especially when our initial estimate $\mathbf{x}_{0|t}$ is founded on substantive prior knowledge. The regularization parameter $\lambda$ offers a balance between these two objectives, ensuring our new estimate aligns with observations while respecting our initial belief encapsulated in $\mathbf{x}_{0|t}$. Note that $\hat{\mathbf{x}}_{0|t}$ usually has a solution in closed form. For MRI reconstruction, the details can be found in appendix A.3.

Secondly, as previously noted, different choices of $\bar{\mathbf{x}}$ result in estimates that are all equally consistent, and the choice of $\mathbf{x}_{0|t}$ represents just one specific solution among the possibilities. We postulate

that the chosen $\bar{\mathbf{x}}$ can profoundly influence the trajectory of the projections. By strategically choosing $\bar{\mathbf{x}}$, we can make our solutions more efficient and accurate, yet ensuring that they respect the desired distribution $q(\mathbf{x})$. In a similar reasoning, the effectiveness of the proximity term in Eq. (10) highly relies on the quality of the prior $\mathbf{x}_{0|t}$. If the prior is not a desirable estimate, it might mislead the optimization. To identify a solution, we return to the posterior mean of $\mathbf{x}_0$ given $\mathbf{x}_t$ discussed in Section 2.2. For Variance Preserving SDE (VPSDEs), the posterior mean is driven based on Tweedie's formula as

$$\mathbf{x}_{0|t} = \mathbb{E}[\mathbf{x}_0|\mathbf{x}_t] = \frac{1}{\sqrt{\bar{\alpha}_t}}\big(\mathbf{x}_t + (1 - \bar{\alpha}_t)\nabla_{\mathbf{x}_t}\log p(\mathbf{x}_t)\big). \tag{11}$$

Ravula et al. (2023) extended Tweedie's formula with an additional measurement $\mathbf{y}$ for Variance Exploding SDE (VESDEs). The updated formula for the conditional posterior mean in VPSDEs (see Appendix A.4.1), can also be presented as

$$\tilde{\mathbf{x}}_{0|t} := \mathbb{E}[\mathbf{x}_0|\mathbf{x}_t, \mathbf{y}] = \frac{1}{\sqrt{\bar{\alpha}_t}}\big(\mathbf{x}_t + (1 - \bar{\alpha}_t)\nabla_{\mathbf{x}_t}\log p(\mathbf{x}_t|\mathbf{y})\big). \tag{12}$$

This new estimation for the initial unconditional prediction functions as an *inner-level* guidance for our method. Hence, we call our *bi-level* guidance strategy, Meta-Guided Diffusion Models (MGDM). Given the relation in Eq. (5), it becomes clear that by integrating the prior score with the likelihood score, we can procure a more precise estimate of $\mathbf{x}_{0|t}$ than by solely relying on the prior. Also, in DPS framework (Chung et al., 2022a), the time-dependent likelihood score is approximated as $\nabla_{\mathbf{x}_t}\log p(\mathbf{y}|\mathbf{x}_t) \simeq \nabla_{\mathbf{x}_t}\log p(\mathbf{y}|\mathbf{x}_{0|t})$. For the scenario where the measurement noise is Gaussian, i.e., $\mathbf{y} \sim \mathcal{N}(\mathbf{y}; \mathcal{A}(\mathbf{x}_0), \boldsymbol{\sigma}_{\mathbf{y}}^2\mathbf{I})$, we then have $\nabla_{\mathbf{x}_t}\log p_t(\mathbf{y}|\mathbf{x}_t) \simeq -1/\sigma_{\mathbf{y}}^2\nabla_{\mathbf{x}_t}\|\mathbf{y} - \mathcal{A}(\mathbf{x}_{0|t})\|_2^2$. In practice, it is assumed that $p_t(\mathbf{y}|\mathbf{x}_{0|t}) \sim \mathcal{N}(\mathbf{y}; \mathcal{A}\mathbf{x}_{0|t}, \sigma_t^2\mathbf{I})$. Building on DPS's result, an approximation of the expectation in Eq. (12) can be established (see Appendix A.4.2) as

$$\tilde{\mathbf{x}}_{0|t} \simeq \frac{1}{\sqrt{\bar{\alpha}_t}}\Big[\mathbf{x}_t - \sqrt{1 - \bar{\alpha}_t}\boldsymbol{\epsilon}_\theta(\mathbf{x}_t, t) - \zeta\nabla_{\mathbf{x}_t}\|\mathbf{y} - \mathcal{A}\mathbf{x}_{0|t}\|_2^2\Big], \tag{13}$$

where $\zeta$ is a likelihood step size. For sampling $\mathbf{x}_{t-1}$, we employ DDIM, one of the most recognized accelerated diffusion sampling methods. This method transitions the stochastic ancestral sampling of DDPM to deterministic sampling, thereby expediting the sampling process.

In addition to the aforementioned procedures, we have implemented a further step termed the 'discrepancy gradient', aiming to refine the recovered samples. This step updates samples by subtracting it from the *gradient* of the squared norm of the *discrepancy* between the optimized estimate $\hat{\mathbf{x}}_{0|t}$ and the initial prediction $\tilde{\mathbf{x}}_{0|t}$ formulated as

$$\hat{\mathbf{x}}_{t-1} = \mathbf{x}_{t-1} - \rho\nabla_{\mathbf{x}_t}\|\hat{\mathbf{x}}_{0|t} - \tilde{\mathbf{x}}_{0|t}\|_2^2, \tag{14}$$

where $\rho$ is the step size. Discrepancy gradient guides $\mathbf{x}_{t-1}$ towards an equilibrium between two values $\hat{\mathbf{x}}_{0|t}$ and $\tilde{\mathbf{x}}_{0|t}$. Our high-level interpretation of this step is that it aids in improving the accuracy of the approximated measurement-conditioned posterior mean $\tilde{\mathbf{x}}_{0|t}$ and reduces the necessity of the proximal optimization step in Alg 2 (line 9).

From the discussion presented above, we summarized the steps of our proposed method in Algorithm 2. We also provided a schematic illustration of the geometrical differences between our MGDM guidance strategy and other SOTA guidance techniques in Figure 1.

## 4 EXPERIMENTS

In this section, we first present the experimental setup, then provide the results, wherein we quantitatively and qualitatively compare our model with the state-of-the-art (SOTA) methods, followed by the ablation study discussed in the last subsection; details on implementation can be found in Appendix A.5.

### 4.1 DATA SETS

To demonstrate the performance of our proposed method, we present our sampling evaluation on three publicly available datasets. For undersampled MRI experiments, we rely on real-valued Brain

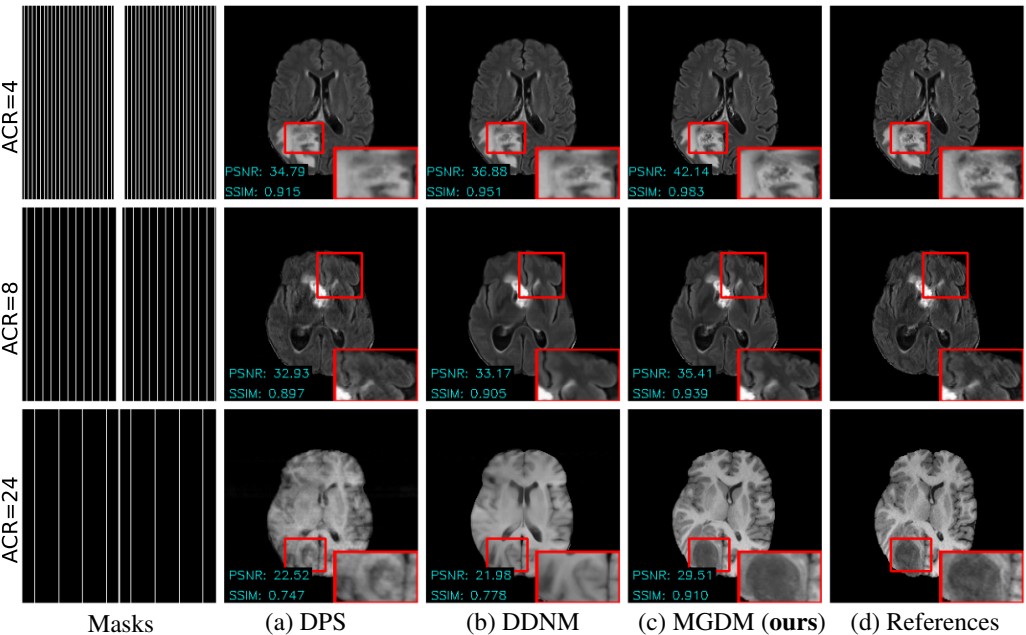

Figure 2: The qualitative results of undersampled MRI reconstruction on the BraTS dataset, depicted for Acceleration Rates (ACR) 4, 8, and 24.

Tumor Segmentation (BraTS) 2021 (Menze et al., 2014; Bakas et al., 2017) and complex-valued fastMRI knee datasets (Zbontar et al., 2018). In our evaluation with the BraTS dataset, we follow the approach outlined in (Song et al., 2021), where 3D MRI volumes are sliced to obtain 297,270 images with a resolution of $240 \times 240$ for the training set. We simulate MRI measurements using the Fast Fourier Transform (FFT) and undersample the k-space using an equispaced Cartesian mask, from an acceleration factor of 4 to 24. When conducting experiments on fastMRI, we follow (Chung & Ye, 2022) to appropriately crop the raw k-space data to $320 \times 320$ pixels. We then generate single-coil minimum variance unbiased estimator (MVUE) images as our ground truth references. The measurements of these images are derived from the fully sampled k-space data multiplied by sensitivity maps computed through the ESPIRiT (Uecker et al., 2014) algorithm. To simulate measurements for fastMRI, the data is processed using the FFT and then undersampled with a one-dimensional Gaussian mask acceleration factor 4 and 8. For the sparse-view CT reconstruction experiment, we used the Lung Image Database Consortium (LIDC) dataset (Armato III et al., 2011; Clark et al., 2013). From this dataset, we derived 130,304 two-dimensional images with a resolution of $320 \times 320$ by slicing the original 3D CT volumes. We produce simulated CT measurements (sinograms), using a parallel-beam setup and evenly spaced 10 and 23 projection angles over 180 degrees to simulate sparse-view acquisition.

## 4.2 BASELINES

We primarily compare our proposed method with two state-of-the-art zero-shot inverse problem solvers: DPS (Chung et al., 2022a) and DDNM (Wang et al., 2022). For the Knee fastMRI dataset, we reported the result of Score-MRI (Chung & Ye, 2022) directly from their paper. To ensure a fair comparison, we adopt the incorporation strategies from these methods, along with appropriate parameter settings within our architecture. Also, for CT reconstruction, we replaced the DPS method with Song's method (ScoreMed) (Song et al., 2021). In our experiments, it was observed that the recurrent use of Filtered Back Projection (FBP) tends to be numerically unstable in DPS, frequently resulting in overflow. This has also been reported by (Chung et al., 2022b). For all experiments, results are reported in terms of peak signal-to-noise ratio (PSNR) and structural similarity (SSIM) metrics on a dataset of 1,000 test images. To further validate the performance of our approach, we provide a quantitative comparison with SOTA-supervised methods. These comparisons are detailed in Appendix A.6.

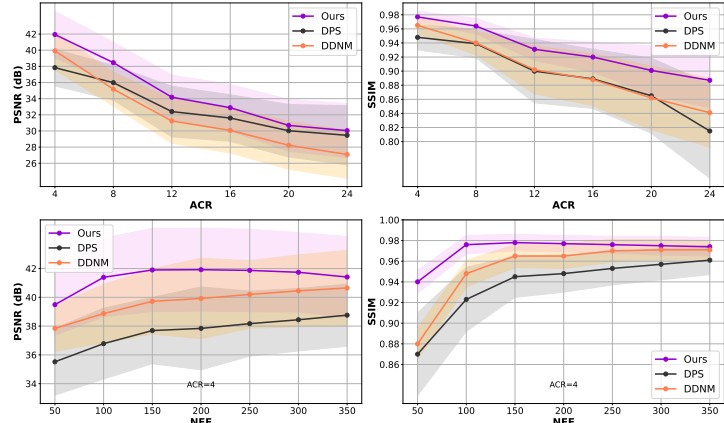

Figure 3: In the array of graphs, the upper row illustrates the undersampled MRI reconstruction results for 200 timesteps at various acceleration rates (ACR), and the lower row displays the results over a span of 350 timesteps at a fixed acceleration rate of 4.

Table 1: Results for undersampled MRI reconstruction on complexed-valued fastMRI Knee dataset.

| Method | 4× ACR | | 8× ACR | |
|---|---|---|---|---|
| | PSNR↑ | SSIM↑ | PSNR↑ | SSIM↑ |
| DPS (Chung et al., 2022a) | $22.41_{\pm3.33}$ | $0.650_{\pm0.080}$ | $21.87_{\pm2.91}$ | $0.607_{\pm0.076}$ |
| DDNM (Wang et al., 2022) | $35.87_{\pm2.68}$ | $0.873_{\pm0.065}$ | $34.04_{\pm2.70}$ | $0.847_{\pm0.071}$ |
| Score-MRI (Chung & Ye, 2022) | $33.96_{\pm1.27}$ | $0.858_{\pm0.028}$ | $30.82_{\pm1.37}$ | $0.762_{\pm0.034}$ |
| MGDM (**ours**) | $\mathbf{36.94}_{\pm2.70}$ | $\mathbf{0.888}_{\pm0.062}$ | $\mathbf{34.98}_{\pm2.66}$ | $\mathbf{0.856}_{\pm0.070}$ |

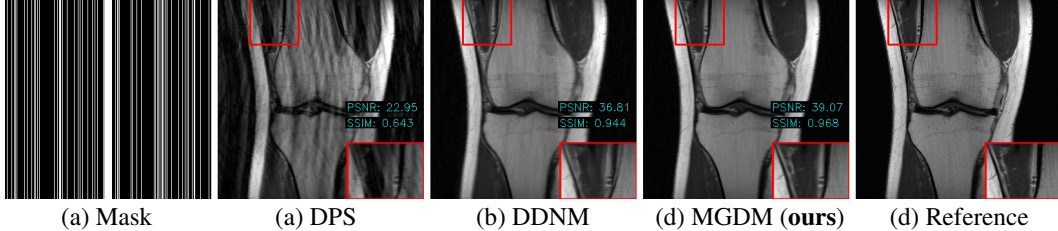

| (a) Mask | (a) DPS | (b) DDNM | (d) MGDM (**ours**) | (d) Reference |
|---|---|---|---|---|

Figure 4: The qualitative representative results of the fastMRI knee dataset at ACR 4 with 100 steps.

## 4.3 RESULTS

In Figure 2, we display the BraTS image reconstruction results using different methods for test measurements undersampled at 4x, 8x, and 24x acceleration factors. Our MGDM method achieves superior image fidelity, preserving lesion heterogeneities at 4x and 8x undersampling levels. Unlike other methods, MGDM maintains data fidelity even at 24x undersampling, producing highly consistent images with the ground truth. More examples can be found in Appendix A.8, showcasing MGDM's noise and motion handling. In Figure 3, a comparative analysis of reconstruction quality is presented, employing metrics such as PSNR and SSIM on a dataset of 1000 BraTS images across a diverse range of Network Function Evaluators (NFEs) and Acceleration Rates (ACR). The evaluation underscores the superior performance of MGDM over other methods, demonstrating not only higher accuracy but also efficiency in computational time. Notably, MGDM, even at a modest 100 NFEs, significantly performs better than other methods operating at a substantially higher 350 NFEs, establishing its noteworthy efficacy in producing accurate reconstructions swiftly. The comparison of various methods on the fastMRI knee dataset with 100 NFEs is presented in Table 1, with an illustrative case showcased in Figure 4. DPS failed to reconstruct acceptable images due to the short 100 sampling steps. Notably, our MGDM method demonstrated superior performance compared to Score-MRI (Chung et al., 2023) and DDNM by a margin of **3dB** and **1dB**, respectively. Figure 5 illustrates the results of reconstructing a CT lung image from 23 projections using multiple methods. Our method recovers finer details, as seen in the zoomed-in views, and achieves the highest PSNR and SSIM values. Table 2 shows the average results from 1000 test CT images using both 23 and 10 projections. Our method slightly outperforms ScoreMed in terms of PSNR and SSIM values, with both significantly surpassing DDNM.

Table 2: Quantitative results of sparse-view CT reconstruction on the LIDC dataset with 350 NFEs.

| Method | 23 projection | | 10 projection | |
|---|---|---|---|---|
| | PSNR↑ | SSIM↑ | PSNR↑ | SSIM↑ |
| FBP | $10.07_{\pm 1.40}$ | $0.218_{\pm 0.070}$ | – | – |
| DDNM (Wang et al., 2022) | $23.76_{\pm 2.21}$ | $0.624_{\pm 0.077}$ | $18.35_{\pm 2.30}$ | $0.696_{\pm 0.047}$ |
| ScoreMed (Song et al., 2021) | $35.24_{\pm 2.71}$ | $0.905_{\pm 0.046}$ | $29.52_{\pm 2.63}$ | $0.823_{\pm 0.061}$ |
| Ours $_{no-r}$ | $25.89_{\pm 2.43}$ | $0.671_{\pm 0.069}$ | $20.14_{\pm 2.35}$ | $0.723_{\pm 0.043}$ |
| MGDM (**ours**) | $\mathbf{35.82}_{\pm 2.45}$ | $\mathbf{0.911}_{\pm 0.052}$ | $\mathbf{30.22}_{\pm 2.48}$ | $\mathbf{0.834}_{\pm 0.056}$ |

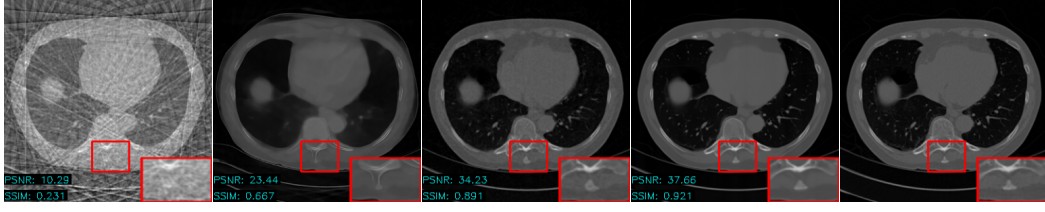

| (a) FBP | (b) DDNM | (b) ScoreMed | (d) MGDM (**ours**) | (e) Reference |
|---|---|---|---|---|

Figure 5: Examples of sparse-view CT reconstruction results on LIDC, all with 23 projections.

Table 3: Ablation study results for undersampled MRI reconstruction using the BraTS dataset.

| Method | 4× ACR | | 8× ACR | | 24× ACR | |
|---|---|---|---|---|---|---|
| | PSNR↑ | SSIM↑ | PSNR↑ | SSIM↑ | PSNR↑ | SSIM↑ |
| DPS (Chung et al., 2022a) | $37.84_{\pm 2.26}$ | $0.948_{\pm 0.018}$ | $35.98_{\pm 2.15}$ | $0.939_{\pm 0.020}$ | $29.46_{\pm 3.66}$ | $0.815_{\pm 0.067}$ |
| DDNM (Wang et al., 2022) | $39.92_{\pm 2.35}$ | $0.965_{\pm 0.012}$ | $35.18_{\pm 2.10}$ | $0.940_{\pm 0.017}$ | $27.09_{\pm 2.94}$ | $0.841_{\pm 0.049}$ |
| Ours $_{no-pr}$ | $32.38_{\pm 1.89}$ | $0.874_{\pm 0.030}$ | $29.56_{\pm 2.01}$ | $0.845_{\pm 0.034}$ | $23.16_{\pm 2.53}$ | $0.794_{\pm 0.044}$ |
| Ours $_{no-ir}$ | $39.97_{\pm 2.31}$ | $0.969_{\pm 0.011}$ | $35.36_{\pm 2.03}$ | $0.943_{\pm 0.015}$ | $27.36_{\pm 2.78}$ | $0.849_{\pm 0.041}$ |
| Ours $_{no-r}$ | $41.54_{\pm 2.90}$ | $\mathbf{0.980}_{\pm 0.008}$ | $38.02_{\pm 2.31}$ | $0.961_{\pm 0.009}$ | $29.87_{\pm 3.31}$ | $0.887_{\pm 0.036}$ |
| Ours $_{no-i}$ | $41.37_{\pm 2.72}$ | $0.967_{\pm 0.009}$ | $37.06_{\pm 2.04}$ | $0.923_{\pm 0.011}$ | $28.37_{\pm 3.23}$ | $0.832_{\pm 0.047}$ |
| MGDM (**ours**) | $\mathbf{41.94}_{\pm 2.88}$ | $0.977_{\pm 0.008}$ | $\mathbf{38.46}_{\pm 2.54}$ | $\mathbf{0.964}_{\pm 0.011}$ | $\mathbf{30.04}_{\pm 3.33}$ | $\mathbf{0.887}_{\pm 0.039}$ |

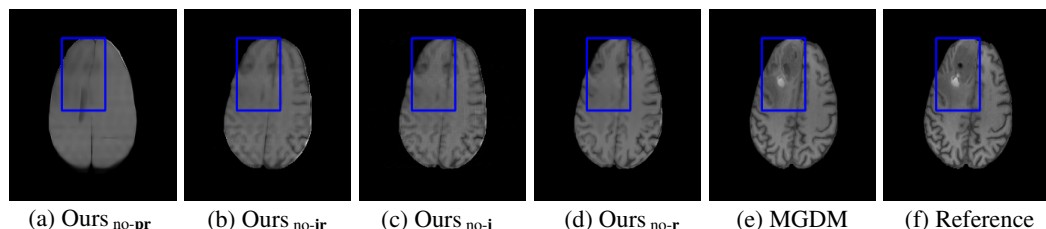

| (a) Ours $_{no-pr}$ | (b) Ours $_{no-ir}$ | (c) Ours $_{no-i}$ | (d) Ours $_{no-r}$ | (e) MGDM | (f) Reference |
|---|---|---|---|---|---|

Figure 6: A representative visual result of the ablation study, showcasing the 24x scenario.

## 4.4 ABLATION STUDIES

To assess the impact of key components in our sampling algorithm, we performed ablations on the undersampled MRI task using the BraTS dataset. The summarized outcomes are presented in Table 3, evaluating four key variations in Algorithm 2: (i) the exclusion of **p**roximal optimization (step **9**) and **r**efinement (step **11**) termed 'no-**pr**', (ii) the omission of **i**nitial prediction (step **8**) and **r**efinement (step **11**) designated as 'no-**ir**', (iii) the absence of **i**nitial prediction alone (step **8**) noted as 'no-**i**', and (iv) the removal of **r**efinement alone (step **11**) referred to as 'no-**r**'. Our observations indicate that proximal optimization plays the most substantial role in our MGDM method, with improvements achieved through more accurate initial prediction and further refinement. Remarkably, our algorithm outperforms all baselines even without the refinement step, yet further improves performance when this step is incorporated, as illustrated in Fig 6 (see d and e).

## 5 CONCLUSION

In this paper, we propose an effective approach for tackling inverse problems in medical imaging. Through extensive experiments, our method demonstrates its superiority to other methods on several highly heterogeneous, publicly available medical datasets, thereby validating our analysis. Theoretically, our approach is amenable to resolving other linear inverse problems such as inpainting, super-resolution, deblurring, and so forth, provided that the pertinent diffusion model is accessible. The limitations of this study and future work are discussed in Appendix A.7.

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

# A APPENDIX

## A.1 RELATED WORKS

A solution to the inverse problem $\mathbf{y} = \mathcal{A}\mathbf{x} + \mathbf{n}$, can be probabilistically derived via the maximum likelihood estimation (MLE), defined as $\mathbf{x}_{\text{ML}} = \arg\max_{\mathbf{x}} \log p(\mathbf{y}|\mathbf{x})$, where $p(\mathbf{y}|\mathbf{x}) := \mathcal{N}(\mathcal{A}\mathbf{x}, \sigma_{\mathbf{y}}^2)$ represents the likelihood of observation $\mathbf{y}$, ensuring data consistency. Nevertheless, if the forward operator $\mathcal{A}$ is singular, e.g., when $m < n$, the problem is ill-posed. In such cases, it is fundamentally infeasible to uniquely recover the signal set $\mathcal{X}$ using only the observed measurements $\mathcal{Y}$, even in the noiseless scenario where $\mathcal{Y} = \mathcal{A}\mathcal{X}$. This challenge arises due to the nontrivial nature of the null space of $\mathcal{A}$. To mitigate the ill-posedness, it is therefore essential to incorporate an additional assumption based on *prior* knowledge to constrain the space of possible solutions. A predominantly adopted framework that offers a more meaningful solution is Maximum a Posteriori (MAP) estimation which is formulated as $\mathbf{x}_{\text{MAP}} = \arg\max_{\mathbf{x}}[\log p(\mathbf{y}|\mathbf{x}) + \log p(\mathbf{x})]$, where the term $\log p(\mathbf{x})$ encapsulates the prior information of the clean image $\mathbf{x}$.

The concept of priors in solving inverse problems has evolved considerably over time. Classically, many methodologies relied on hand-crafted priors, which are analytically defined constraints such as sparsity (Candès & Wakin, 2008; Tang et al., 2009), low-rank (Fazel et al., 2008; Cui et al., 2014), total variation (Candès et al., 2006), to name but a few, to enhance reconstruction. With the advent of deep learning models, priors have transitioned to being data-driven, yielding significant gains in reconstruction quality (Bora et al., 2017; Mardani et al., 2018; Ardizzone et al., 2018; Goh et al., 2019; Asim et al., 2020; Whang et al., 2021). These priors, whether learned in a supervised or unsupervised fashion, have been integrated within the MAP framework to address ill-posed inverse problems. In the supervised paradigm, the reliance on the availability of paired original images and observed measurements also can potentially limit the model's generalizability. As a result, the trend has shifted towards an increasing interest in unsupervised approaches, where priors are learned implicitly or explicitly using deep generative models. The strategies within the unsupervised learning paradigm vary based on how the learned priors (a.k.a. generative priors) are imposed. For instance, generators $\mathcal{G}_\theta$ in pre-trained generative models such as Generative Adversarial Networks (GANs)(Goodfellow et al., 2016; Bora et al., 2017), Variational Autoencoders (VAEs) (Ardizzone et al., 2018), and Normalizing Flows (NFs) (Asim et al., 2020), are employed as priors to identify the latent code that explains the measurements, as described by the optimization problem $\hat{\mathbf{z}} = \arg\max_{\mathbf{z}} \log p(\mathbf{y}|\mathcal{G}_\theta(\mathbf{z})) + \log p(\mathbf{z})$. In such a way, the solution $\hat{\mathbf{z}}$ is constrained to be within the domain of the generative model. This approach, however, suffers from critical restrictions. In the first place, the low dimensionality of the latent space is a major concern, as it hampers the reconstruction of images that lie outside their manifold. Additionally, it demands computationally expensive iterative updates, given the complexity of generator $\mathcal{G}_\theta$. Crucially, the deterministic nature of the recovered solutions hinders the assessment of the reliability of reconstruction. In fact, MAP inference fails to fully capture the entire range of the solution spectrum, particularly when solving an ill-posed problem that might hold multiple solutions aligned closely with both the observed measurements and prior assumptions.

To account for the variety within the solution domain and to measure reconstruction certainty, the inverse problem is tackled from a Bayesian inference standpoint. Bayesian inference yields a posterior distribution, $p(\mathbf{x}|\mathbf{y})$, from which multiple conditional samples can be extracted (Brooks et al., 2011; Blei et al., 2017). Recently, pre-trained diffusion models (Ho et al., 2020; Nichol & Dhariwal, 2021) are utilized as a powerful generative prior (a.k.a denoiser), in a zero-shot manner, to effectively sample from the conditional posterior (Kadkhodaie & Simoncelli, 2021; Daras et al., 2022; Rombach et al., 2022). The strategies for posterior (conditional) sampling via diffusion models fall into two distinct approaches. In the first approach, diffusion models are trained conditionally, directly embedding the conditioning information $\mathbf{y}$ during the training phase (Ho et al., 2020; Rombach et al., 2022; Liu et al., 2023). However, conditional training tends to require: (i) the assembly of a massive amount of paired data and its corresponding conditioning $(\mathbf{x}, \mathbf{y})$, and (ii) retraining when testing on new conditioning tasks, highlighting the adaptability issues. In the second approach, unconditionally pre-trained diffusion models are employed as generative prior (a.k.a denoiser) to perform conditional sampling for certain tasks. A primary difficulty, however, is how to impose data consistency between measurements and the generated images in each iteration (Chung et al., 2022a; Wang et al., 2022; Chung et al., 2023).

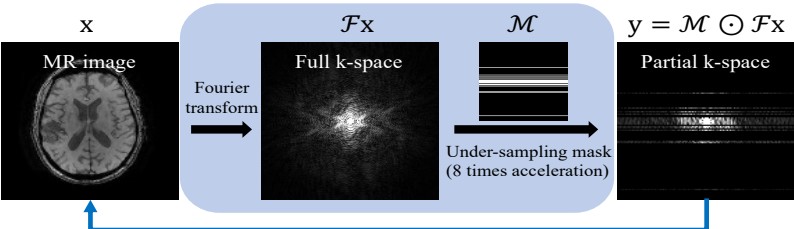

MRI under-sampling reconstruction: $p(\mathbf{x}|\mathbf{y})$

Figure 7: Linear measurement processes for undersampled MR

## A.2 Proof of Proposition 3.1

Consider an iteration of gradient descent, initialized from $\mathbf{x}^{(0)}$, on the least squares problem

$$\mathbf{x}^{(t+1)} = \mathbf{x}^{(t)} + \alpha \mathcal{A}^T (\mathbf{y} - \mathcal{A}\mathbf{x}^{(t)}).$$

Defining $\mathbf{r}^{(t)} = \mathbf{y} - \mathcal{A}\mathbf{x}^{(t)}$, it follows that

$$\mathbf{r}^{(t+1)} = \left(\mathbf{I} - \alpha \mathcal{A}\mathcal{A}^T\right) \mathbf{r}^{(t)} = \ldots = \left(\mathbf{I} - \alpha \mathcal{A}\mathcal{A}^T\right)^{t+1} \mathbf{r}^{(0)}.$$

Hence,

$$\mathbf{x}^{(t+1)} = \mathbf{x}^{(t)} + \alpha \mathcal{A}^T \left(\mathbf{I} - \alpha \mathcal{A}\mathcal{A}^T\right)^t \mathbf{r}^{(0)}$$

$$= \mathbf{x}^{(0)} + \alpha \mathcal{A}^T \sum_{i=0}^{t} \left(\mathbf{I} - \alpha \mathcal{A}\mathcal{A}^T\right)^i \mathbf{r}^{(0)}$$

$$= \mathbf{x}^{(0)} + \alpha \sum_{i=0}^{t} \left(\mathbf{I} - \alpha \mathcal{A}^T\mathcal{A}\right)^i \mathcal{A}^T \mathbf{r}^{(0)}.$$

Subsequently, as long as $0 < \alpha < 1/\|\mathcal{A}\|^2$, from (Ben-Israel & Charnes, 1963, Theorem 16), we get

$$\lim_{t \to \infty} \mathbf{x}^{(t)} = \mathbf{x}^{(0)} + \alpha \sum_{i=0}^{\infty} \left(\mathbf{I} - \alpha \mathcal{A}^T\mathcal{A}\right)^i \mathcal{A}^T \mathbf{r}^{(0)} = \mathbf{x}^{(0)} + \mathcal{A}^\dagger \mathbf{r}^{(0)}.$$

## A.3 Closed-form solutions

Consider the following optimization problem in Eq. (10)

$$\hat{\mathbf{x}}_{0|t} = \arg\min_{\mathbf{x}} \frac{1}{2} \|\mathbf{y} - \mathcal{A}\mathbf{x}\|_2^2 + \frac{\lambda}{2} \|\mathbf{x} - \mathbf{x}_{0|t}\|_2^2.$$

For the MRI reconstruction task, we express $\mathcal{A}\mathbf{x} = \mathcal{M} \odot (\mathcal{F}\mathbf{x}) = \mathcal{M} \odot \mathbf{w}$, where $\mathcal{M}$ represents the Cartesian equispaced mask, $\mathcal{F}$ is the Fourier matrix, and $\odot$ signifies element-wise multiplication. The whole process and matrix decomposition for MRI is illustrated in Fig 7. Given this definition, and considering the identity $\arg\min_{\mathbf{x}}\|\mathbf{x} - \mathbf{x}_{0|t}\|_2^2 = \arg\min_{\mathbf{x}}\|\mathcal{F}\mathbf{x} - \mathcal{F}\mathbf{x}_{0|t}\|_2^2$, then the optimization problem in terms of $\mathbf{w}$ can be redefined as

$$\hat{\mathbf{w}}_{0|t} = \arg\min_{\mathbf{w}} \frac{1}{2} \|\mathcal{M} \odot \mathbf{w} - \mathbf{y}\|_2^2 + \frac{\lambda}{2} \|\mathbf{w} - \mathbf{w}_{0|t}\|_2^2.$$

By expanding the L2-norm terms, we obtain

$$\hat{\mathbf{w}}_{0|t} = \arg\min_{\mathbf{w}} \sum_{i=1}^{n} (m_i w_i - y_i)^2 + \lambda \sum_{i=1}^{n} (w_i - w_{0|t}^i)^2.$$

The solution for $\hat{\mathbf{w}}_{0|t}$ is

$$\hat{\mathbf{w}}_{0|t} = \frac{\mathcal{M}\mathbf{y} + \lambda\mathbf{w}_{0|t}}{\mathcal{M} + \lambda}.$$

Given the relation $\hat{\mathbf{x}}_{0|t} = \mathcal{F}^{-1}\hat{\mathbf{w}}_{0|t}$, we can then deduce

$$\boxed{\hat{\mathbf{x}}_{0|t} = \mathcal{F}^{-1}\left(\frac{\mathcal{M}\mathbf{y} + \lambda\mathcal{F}\mathbf{x}_{0|t}}{\mathcal{M} + \lambda}\right)}$$

Consider the following range-null space decomposition defined in Eq. (8)

$$\hat{\mathbf{x}}_{0|t} = \mathcal{A}^{\dagger}\mathbf{y} + \left(\mathbf{I} - \mathcal{A}^{\dagger}\mathcal{A}\right)\mathbf{x}_{0|t}.$$

where $\mathcal{A}^{\dagger}$ denotes the pseudo-inverse of matrix $\mathcal{A}$ and $\mathbf{I}$ is the identity matrix.

For MRI, the forward operator is modelled as $\mathcal{A} = \mathcal{M}\mathcal{F}$. An important property that arises is $\mathcal{A}\mathcal{A}\mathcal{A} \equiv \mathcal{A}$, which suggests that $\mathcal{A}$ itself can be represented as its pseudo-inverse $\mathcal{A}^{\dagger}$. With this property, the pseudo-inverse is then expressed as $\mathcal{A}^{\dagger} = \mathcal{F}^{-1}\mathcal{M}$. Substituting this representation into our original expression, we obtain

$$\hat{\mathbf{x}}_{0|t} = \mathcal{F}^{-1}\mathcal{M}\mathbf{y} + \left(\mathbf{I} - \mathcal{F}^{-1}\mathcal{M}\mathcal{F}\right)\mathbf{x}_{0|t}.$$

Using the Fourier identity $\mathcal{F}^{-1}\mathcal{F} = \mathbf{I}$, we can further simplify this to:

$$\boxed{\hat{\mathbf{x}}_{0|t} = \mathcal{F}^{-1}\left(\mathcal{M}\mathbf{y} + (\mathbf{I} - \mathcal{M})\mathcal{F}\mathbf{x}_{0|t}\right)}$$

### A.4 POSTERIOR MEAN

#### A.4.1 POSTERIOR MEAN WITH ADDITIONAL MEASUREMENT FOR VPSDE

A notable SDE with an analytic transition probability is the variance-preserving SDE (VPSDE) (Song et al., 2020b; Karras et al., 2022), which considers $\mathbf{f}(\mathbf{x}_t, t) = -\frac{1}{2}\beta(t)\mathbf{x}_t$ and $g(t) = \sqrt{\beta(t)}$, where $\beta(t) = \beta_{min} + t(\beta_{max} - \beta_{min})$; and its transition probability follows a Gaussian distribution of $p_{0t}(\mathbf{x}_t|\mathbf{x}_0) = \mathcal{N}(\mathbf{x}_t; \boldsymbol{\mu}_t\mathbf{x}_0, \boldsymbol{\sigma}_t^2\mathbf{I})$ with $\boldsymbol{\mu}_t = \exp\{-\frac{1}{2}\int_0^t \beta(s)s\}$ and $\boldsymbol{\sigma}_t^2 = 1 - \exp\{-\int_0^t \beta(s)s\}$. Given such transition probability, we seek to derive the corresponding posterior mean with additional measurement.

Begin by representing the distribution $p_t(\mathbf{x}_t|\mathbf{y})$ as marginalizing out $\mathbf{x}_0$ conditioned on $\mathbf{y}$:

$$p_t(\mathbf{x}_t|\mathbf{y}) = \int_{\mathbf{x}_0} p_t(\mathbf{x}_t|\mathbf{x}_0, \mathbf{y})p_0(\mathbf{x}_0|\mathbf{y})d\mathbf{x}_0.$$

Differentiate w.r.t. $\mathbf{x}_t$ on both sides

$$\nabla_{\mathbf{x}_t}p_t(\mathbf{x}_t|\mathbf{y}) = \int_{\mathbf{x}_0} p_0(\mathbf{x}_0|\mathbf{y})\nabla_{\mathbf{x}_t}p_t(\mathbf{x}_t|\mathbf{x}_0, \mathbf{y})d\mathbf{x}_0.$$

With our new probability distribution model, the gradient becomes

$$\nabla_{\mathbf{x}_t}\log p_t(\mathbf{x}_t|\mathbf{x}_0) = \frac{(\boldsymbol{\mu}_t\mathbf{x}_0 - \mathbf{x}_t)}{\boldsymbol{\sigma}_t^2}.$$

Inserting this into our previous equation, we have

$$\nabla_{\mathbf{x}_t}p_t(\mathbf{x}_t|\mathbf{y}) = \int_{\mathbf{x}_0} p_0(\mathbf{x}_0|\mathbf{y})p_t(\mathbf{x}_t|\mathbf{x}_0, \mathbf{y})\frac{(\boldsymbol{\mu}_t\mathbf{x}_0 - \mathbf{x}_t)}{\boldsymbol{\sigma}_t^2}d\mathbf{x}_0.$$

Simplifying the above equation, we get:

$$\nabla_{\mathbf{x}_t}p_t(\mathbf{x}_t|\mathbf{y}) = \frac{1}{\boldsymbol{\sigma}_t^2}\left[\int_{\mathbf{x}_0} p_0(\mathbf{x}_0|\mathbf{y})p_t(\mathbf{x}_t|\mathbf{x}_0, \mathbf{y})\boldsymbol{\mu}_t\mathbf{x}_0 d\mathbf{x}_0 - \int_{\mathbf{x}_0} p_0(\mathbf{x}_0|\mathbf{y})p_t(\mathbf{x}_t|\mathbf{x}_0, \mathbf{y})\mathbf{x}_t d\mathbf{x}_0\right].$$

Using Bayes' rule and recognizing the marginalization, we get:

$$\nabla_{\mathbf{x}_t} p_t(\mathbf{x}_t|\mathbf{y}) = \frac{1}{\boldsymbol{\sigma}_t^2} \left[ \int_{\mathbf{x}_0} \boldsymbol{\mu}_t \mathbf{x}_0 p_t(\mathbf{x}_t|\mathbf{y}) p_0(\mathbf{x}_0|\mathbf{x}_t,\mathbf{y}) d\mathbf{x}_0 - \mathbf{x}_t p_t(\mathbf{x}_t|\mathbf{y}) \right].$$

$$\nabla_{\mathbf{x}_t} p_t(\mathbf{x}_t|\mathbf{y}) = \frac{1}{\boldsymbol{\sigma}_t^2} \left[ \boldsymbol{\mu}_t p_t(\mathbf{x}_t|\mathbf{y}) \mathbb{E}[\mathbf{x}_0|\mathbf{x}_t,\mathbf{y}] - \mathbf{x}_t p_t(\mathbf{x}_t|\mathbf{y}) \right].$$

$$\frac{\nabla_{\mathbf{x}_t} p_t(\mathbf{x}_t|\mathbf{y})}{p_t(\mathbf{x}_t|\mathbf{y})} = \frac{1}{\boldsymbol{\sigma}_t^2} \left[ \boldsymbol{\mu}_t \mathbb{E}[\mathbf{x}_0|\mathbf{x}_t,\mathbf{y}] - \mathbf{x}_t) \right].$$

Using the identity property of logarithm $\nabla_{\mathbf{x}} \log p(\mathbf{x}) = \nabla_{\mathbf{x}} p(\mathbf{x})/p(\mathbf{x})$, we can rewrite:

$$\nabla_{\mathbf{x}_t} \log p_t(\mathbf{x}_t|\mathbf{y}) = \frac{1}{\boldsymbol{\sigma}_t^2} \left[ \boldsymbol{\mu}_t \mathbb{E}[\mathbf{x}_0|\mathbf{x}_t,\mathbf{y}] - \mathbf{x}_t \right].$$

From this, the posterior mean becomes:

$$\mathbb{E}[\mathbf{x}_0|\mathbf{x}_t,\mathbf{y}] = \frac{\mathbf{x}_t + \boldsymbol{\sigma}_t^2 \nabla_{\mathbf{x}_t} \log p_t(\mathbf{x}_t|\mathbf{y})}{\boldsymbol{\mu}_t}.$$

This shows that the posterior mean of $\mathbf{x}_0$ conditioned on $\mathbf{x}_t$ and $\mathbf{y}$ now incorporates a scaling by $\boldsymbol{\mu}_t$. By considering $\boldsymbol{\mu}_t = \sqrt{\overline{\alpha}_t}$ and $\boldsymbol{\sigma}_t^2 = 1 - \overline{\alpha}_t$, we have then

$$\mathbb{E}[\mathbf{x}_0|\mathbf{x}_t,\mathbf{y}] = \frac{1}{\sqrt{\overline{\alpha}_t}} (\mathbf{x}_t + (1 - \overline{\alpha}_t)\nabla_{\mathbf{x}_t} \log p_t(\mathbf{x}_t|\mathbf{y})).$$

### A.4.2 APPROXIMATED CONDITIONAL POSTERIOR MEAN

$$\mathbb{E}[\mathbf{x}_0|\mathbf{x}_t,\mathbf{y}] = \frac{1}{\sqrt{\overline{\alpha}_t}} (\mathbf{x}_t + (1 - \overline{\alpha}_t)\nabla_{\mathbf{x}_t} \log p(\mathbf{x}_t|\mathbf{y}))$$

Considering Eq. (5) we have

$$\mathbb{E}[\mathbf{x}_0|\mathbf{x}_t,\mathbf{y}] = \frac{1}{\sqrt{\overline{\alpha}_t}} (\mathbf{x}_t + (1 - \overline{\alpha}_t)(\nabla_{\mathbf{x}_t} \log p(\mathbf{x}_t) + \nabla_{\mathbf{x}_t} \log p(\mathbf{y}|\mathbf{x}_t)))$$

By knowing that $\nabla_{\mathbf{x}_t} \log p(\mathbf{x}_t) \simeq \frac{-1}{\sqrt{1-\overline{\alpha}_t}} \boldsymbol{\epsilon}_\theta(\mathbf{x}_t, t)$, then we get

$$\mathbb{E}[\mathbf{x}_0|\mathbf{x}_t,\mathbf{y}] \simeq \frac{1}{\sqrt{\overline{\alpha}_t}} (\mathbf{x}_t + (1 - \overline{\alpha}_t)(\frac{-1}{\sqrt{1-\overline{\alpha}_t}} \boldsymbol{\epsilon}_\theta(\mathbf{x}_t, t) + \nabla_{\mathbf{x}_t} \log p(\mathbf{y}|\mathbf{x}_t)))$$

which can be simplified further as

$$\mathbb{E}[\mathbf{x}_0|\mathbf{x}_t,\mathbf{y}] \simeq \frac{1}{\sqrt{\overline{\alpha}_t}} (\mathbf{x}_t - \sqrt{1-\overline{\alpha}_t} \boldsymbol{\epsilon}_\theta(\mathbf{x}_t, t) + (1 - \overline{\alpha}_t)\nabla_{\mathbf{x}_t} \log p(\mathbf{y}|\mathbf{x}_t)))$$

From approximation made by DPS (Chung et al., 2022a), that is, $\nabla_{\mathbf{x}_t} \log p_t(\mathbf{y}|\mathbf{x}_t) \simeq -1/\boldsymbol{\sigma}_{\mathbf{y}}^2 \nabla_{\mathbf{x}_t} \|\mathbf{y} - \mathcal{A}(\mathbf{x}_{0|t})\|_2^2$, we then get

$$\tilde{\mathbf{x}}_{0|t} \simeq \frac{1}{\sqrt{\overline{\alpha}_t}} \left[ \mathbf{x}_t - \sqrt{1-\overline{\alpha}_t} \boldsymbol{\epsilon}_\theta(\mathbf{x}_t, t) - \zeta\nabla_{\mathbf{x}_t} \|\mathbf{y} - \mathcal{A}\mathbf{x}_{0|t}\|_2^2 \right],$$

### A.5 IMPLEMENTATION

To learn the prior, we train diffusion-based generative networks using the ADM architecture (Dhariwal & Nichol, 2021) and the default parameters presented by (Song et al., 2021). The models are trained with classifier-free diffusion guidance without dropout probability. We train one network for the real-valued Brats dataset, one for the complexed-value fastMRI dataset, and one for the LIDC-CT dataset.

### A.5.1 Training/Sampling Settings and Hyper-parameters

For a sampling of our method, we fine-tune parameters $(\zeta, \lambda, \rho)$ using cross-validation for each datasets. We observed that extensive hyper-parameter tuning is not required to obtain top-performance results. Accordingly, we limit the hyper-parameter search for each task to $\zeta \in [0, 2]$, $\rho \in [0, 4.5]$, $\lambda \in [1^{-2}, 1^{-3}]$. For other methods of DPS and DDNM, we relied on their implementation. We mostly followed implementation by DDNM (Wang et al., 2022) for the sampling process.

In the BraTS and fastMRI experiments, the parameter $\eta$ is set to 0.85. For the LIDC-CT experiment, $\eta$ is assigned a value of 0.9. For the BraTS dataset, we sample over 200 timesteps; for fastMRI, 100 timesteps; and for the LIDC-CT dataset, 350 timesteps are considered.

### A.6 Comparing MGDM with Supervised Methods

Similar to other zero-shot inverse problem solvers (Chung et al., 2022a; Wang et al., 2022; Kawar et al., 2022), MGDM is superior to existing supervised methods (Zhou & Zhou, 2020; Wei et al., 2020) in these dimensions:

- MGDM can be a zero-shot solver for diverse tasks, while supervised methods need to train separate models for each task and sampling patterns.
- MGDM demonstrates robustness to patterns of undersampling and sparsification, whereas supervised techniques exhibit weak generalizability.
- MGDM, akin to ScoreMed (Song et al., 2021) and Score-MRI (Chung & Ye, 2022), achieves notably enhanced results on medical datasets compared to supervised methods.

These claims are substantiated by the experimental results in Table 4. The results are reported from (Song et al., 2021; Chung & Ye, 2022).

Table 4: Comparison of SSIM and PSNR indicators for different methods across three datasets with different acceleration rates and projections each.

| Method | BraTS-MRI | | | | fastMRI | | | | LIDC-CT | |
| --- | --- | --- | --- | --- | --- | --- | --- | --- | --- | --- |
| | 8× ACR | | 24× ACR | | 4× ACR | | 8× ACR | | 23 **Proj** | |
| | PSNR↑ | SSIM↑ | PSNR↑ | SSIM↑ | PSNR↑ | SSIM↑ | PSNR↑ | SSIM↑ | PSNR↑ | SSIM↑ |
| DuDoRNet (Zhou & Zhou, 2020) | 37.88 | 0.985 | 18.46 | 0.662 | 33.46 | 0.856 | 29.65 | 0.777 | – | – |
| SIN-4c-PRN (Wei et al., 2020) | – | – | – | – | – | – | – | – | 30.48 | 0.895 |
| **MGDM** | **38.46** | 0.964 | **30.04** | **0.887** | **36.94** | **0.888** | **34.98** | **0.856** | **35.82** | **0.911** |

### A.7 Limitations, and Future Work

Several limitations remain that merit further examination.

- Despite achieving superior reconstruction results compared to other methods (Song et al., 2021; Chung et al., 2022a; Wang et al., 2022) and demonstrating more efficient sampling for medical imaging applications (Chung & Ye, 2022; Song et al., 2021; Jalal et al., 2021; Chung et al., 2023), MGDM remains sensitive to hyperparameters. Therefore, exploring a more general hyperparameter tuning approach, such as Bayesian optimization, would be beneficial.
- This study acknowledges that a comprehensive theoretical analysis of the 'Discrepancy Gradient' step within the MGDM algorithm has not been thoroughly explored. While empirical evidence suggests an enhancement in reconstruction results attributable to this step, a gap remains in the theoretical understanding that deserves further investigation.
- It should be noted that our CT simulation adheres to the 2D parallel beam geometry assumption, aligning with the baseline models used in other studies for direct comparison. This differs from the more complex and realistic 3D cone-beam CT or helical CT simulations (Kim et al., 2014). Additionally, the BraTS dataset, employed both in our study and by the baseline methods, has been indicated in a recent paper (Shimron et al., 2022) to have an overestimated undersampling factor, which arises from the conjugate symmetry of k-space inherent in real-valued images.

In future work, we intend to extend our model to work with 3D simulations and in the presence of distributional shifts.

## A.8 ADDITIONAL RESULTS

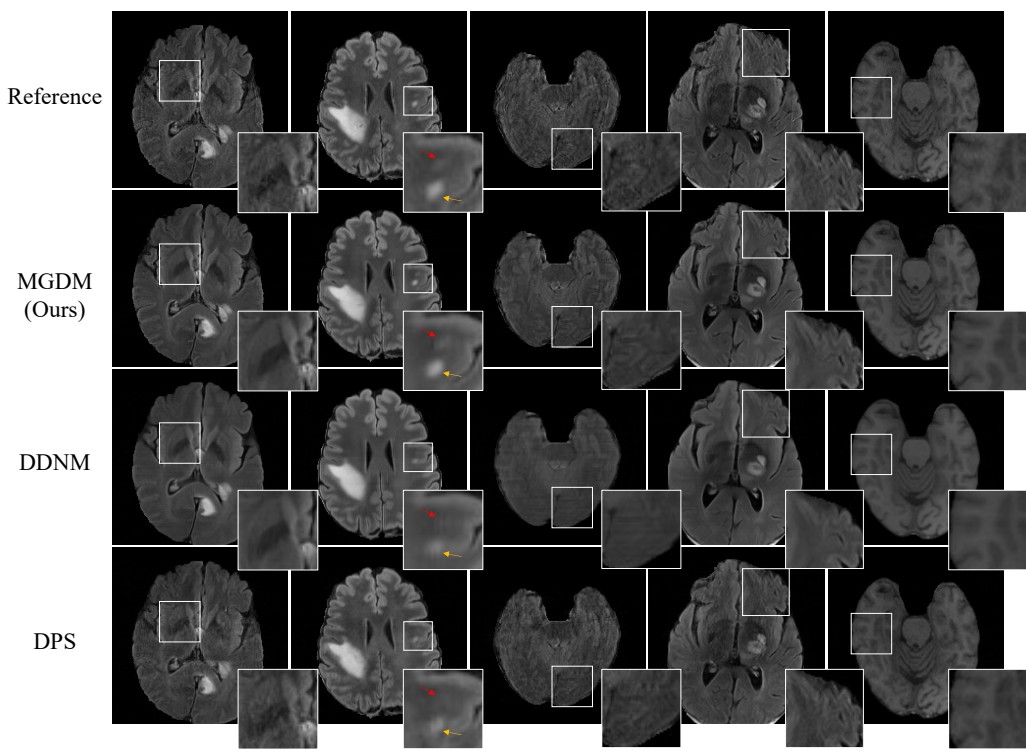

Figure 8: Additional results from undersampled MRI reconstruction on Brats at 8x acceleration rate.

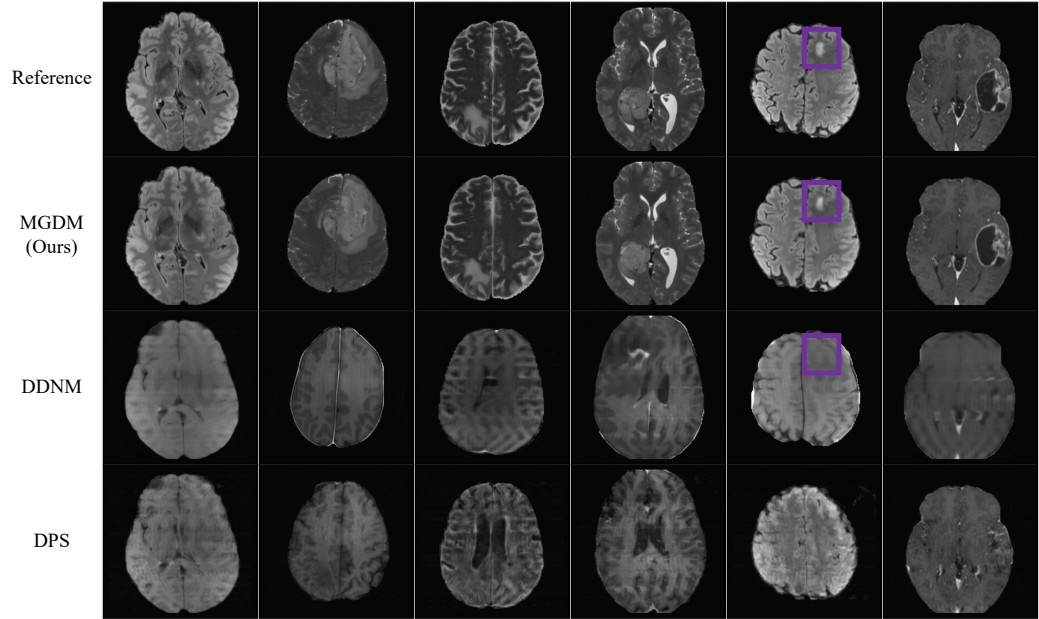

Figure 9: Additional results from undersampled MRI reconstruction on Brats at 24x acceleration rate.

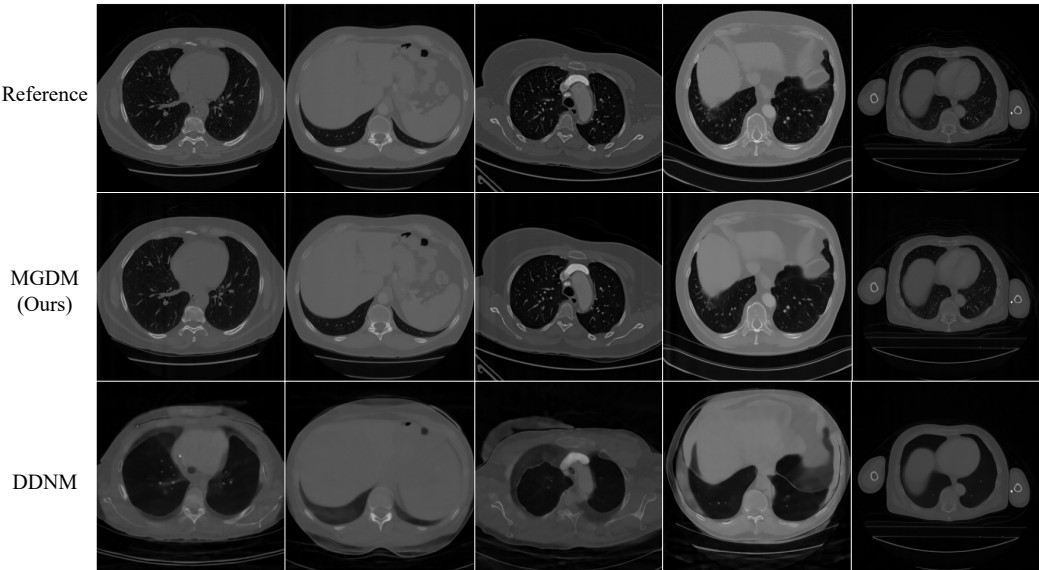

Figure 10: Additional results from sparse-view CT reconstruction on LIDC dataset with 23 projections.

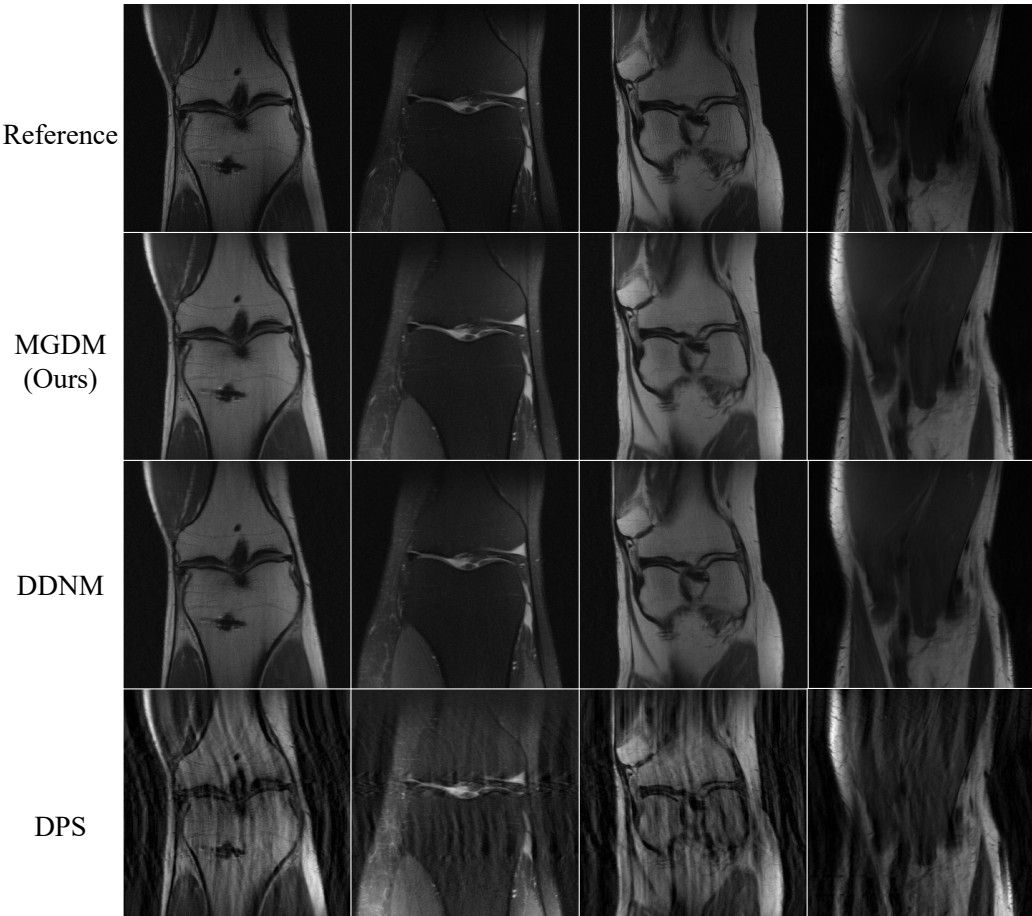

Figure 11: Additional reconstruction results for undersampled knee fastMRI at 4x acceleration rate.

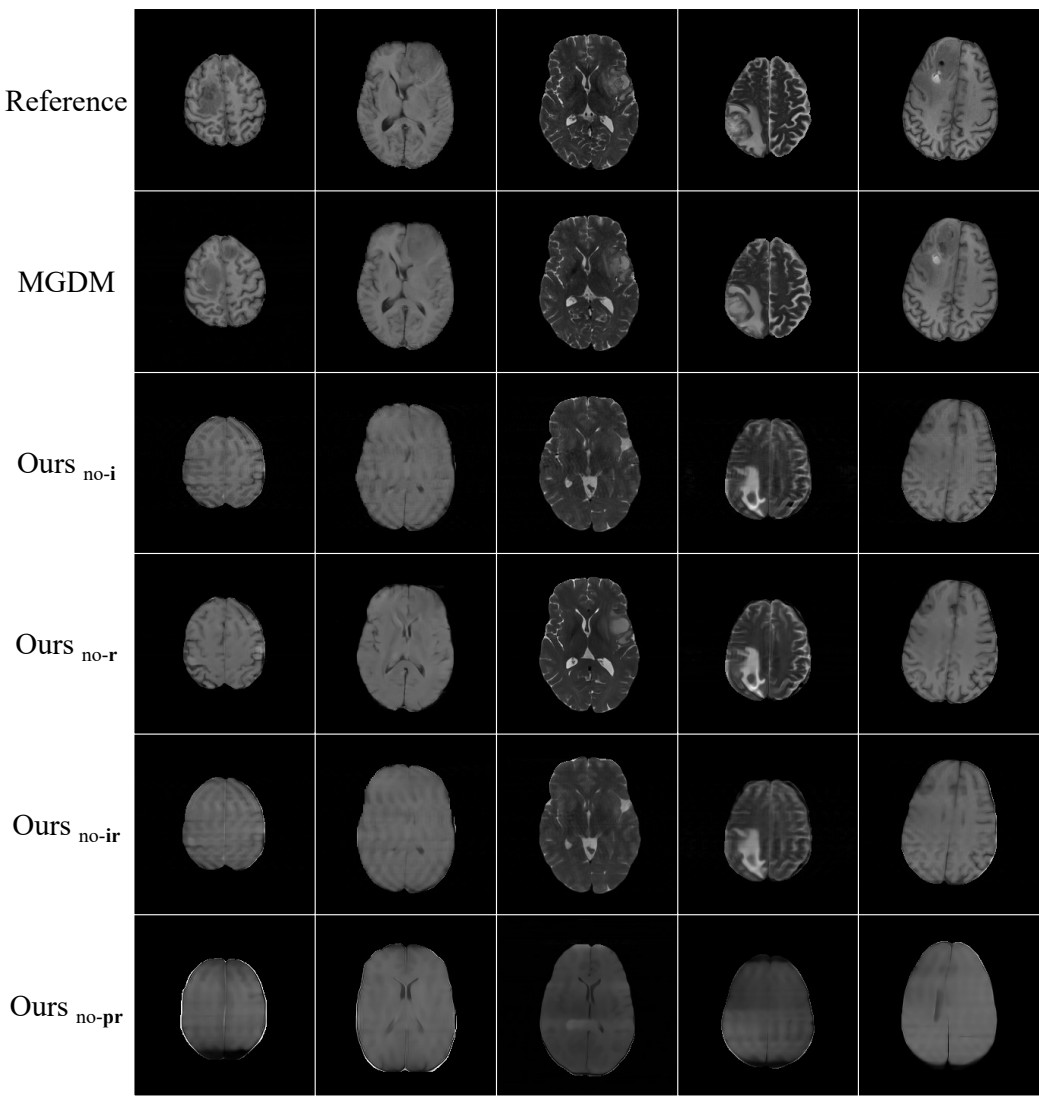

Figure 12: Additional results of our ablation study from undersampled MRI reconstruction on Brats at 24x acceleration rate.

