# OpenReview forum: "Meta-Guided Diffusion Models for Zero-Shot Medical Imaging Inverse Problems"
_ICLR.cc/2024/Conference — ICLR 2024 Conference Withdrawn Submission_

### Official Review · Reviewer_dvv8 · 2023-10-25

**Soundness:** 3 good
**Presentation:** 3 good
**Contribution:** 3 good
**Rating:** 5
**Confidence:** 5

**Summary:**

The paper titled: META-GUIDED DIFFUSION MODELS FOR ZERO-SHOT MEDICAL IMAGING INVERSE PROBLEMS presents a novel sampling strategy (MGDM) for solving inverse problems in medical imaging using diffusion models. Adopting a pre-trained unconditioned diffusion model to conform to the measurement constrain has been an open problem, though there have been quite a few attempts, its still not a well-solved problem. This work introduce an effective bi-level guidance strategy, that acts as a stronger regularizer. The authors evaluated the proposed algorithm on 2 MRI benchmarks and 1 CT benchmark, showing superior performance compared to existing SOTAs.

**Strengths:**

1. This paper introduces a novel yet simple sampling strategy (MGDM) for zero-shot medical imaging inverse problems. I like the core idea of range-null space analysis and using closed-form least square to better conform to the measurements. From the results, the improvements are significant with this simple clean design.

2. The paper is well-written the theory, proof, figures. The core idea is clearly delivered. I truly enjoy reading it, very comprehensive.

3. The authors evaluated the proposed approach on various benchmarks (2 MRI and 1 CT), the ablation studies are well-designed.

**Weaknesses:**

1. Miss the baseline comparisons with supervised method and [Robust compressed sensing MRI with deep generative priors]. I think the current benchmark comparisons are fair, but would curious on how MGDM compare to supervised methods [like MODL: https://arxiv.org/abs/1712.02862] and one of the earlier generative model for MRI recon using Langevin dynamics:  [Robust compressed sensing MRI with deep generative priors].

2. Regarding BraTS dataset: In general, I'm not a big fan of BraTS dataset, since its real-valued images. Therefore, one big problem is that if you perform Fourier Transform, the k-space is conjugate symmetric, and the undersampling factor is not what it is. For example, ACR = 8 effectively represents ACR = 4, this can be miss leading. Please refer and consider citing the paper: https://www.pnas.org/doi/10.1073/pnas.2117203119 Implicit data crimes: Machine learning bias arising from misuse of public data, that discussed this problem. I won't against using BraTS (its a great dataset), but should mention this issue.

Meanwhile, I would appreciate more results on FastMRI dataset, I only see one visual example - Figure 5, without the undersampling pattern, and much descriptions.

3. From results in Figure 2, I am amazed but also confused on the third column ACR=24, despite the effective undersampling rate, with only a few lines, I don't expect the model accurately predicting the tumor. Could you please elaborate on this results? I would like to learn more on what you think?

Minor problems:

1. The figures are not of high-resolution and some of them are compressed. For example, Figure 5, I can actually see jpeg artifacts in all images, which is not acceptable for imaging related papers. Please fix this problem. Also the knee orientation is up-side-down, please fix it.

2. Wrong citation: In the Dataset Section, ESPIRiT is not cited correctly, should be [ESPIRiT — An Eigenvalue Approach to Autocalibrating Parallel MRI: Where SENSE meets GRAPPA], please fix it and check other citations.

**Questions:**

1. Please provide some comprehensive explanation on why MGDM can recover the brain tumor given such a high undersample rate.
2. Could you add some visual results for your ablation studies? I would want to see how the reconstruction results look like with different ablations.

---

> ### Author Response · Authors · 2023-11-22
> **Official Response of Submission7428**
>
> We thank the reviewer for their constructive comments. Below are our point-by-point responses to Weaknesses and Questions:
>
> **Weakness 1:** We acknowledge that “Robust Compressed Sensing MRI” is one of the first pioneer papers in the field, cited in our introduction section. Our decision not to test this method was driven by two primary reasons. Firstly, according to Chung et al. (2022) [1], the approach described in “Robust Compressed Sensing MRI” addresses linear inverse problems by employing an approximation $\nabla_{x_t} \log p_t(y|x) \approx \frac{A^H(y - Ax)}{\sigma^2}$, which is valid when ${n}$ is considered Gaussian noise with variance $\sigma^2$. However, this approximation only holds $t=0$ and fails at other noise levels relevant to the generative process. Secondly, while state-of-the-art (SOTA) methods did not benchmark against this approach, Song's method (Score-Med) [2] did include it as a baseline and outperformed it. Moreover, our results have surpassed those of Song's method.
>
> Regarding the inclusion of supervised methods, we faced uncertainty. The challenge lies in the lack of fair comparative studies, as supervised methods have different architectures and hyper-parameters. Despite that, we have now reported the quantitative comparisons with SOTA supervised methods in Appendix Table 4. Consistent with other zero-shot inverse problem solvers, such as those discussed in Chung et al. (2022) [1], Wang et al. (2022) [3], and Kawar et al. (2022) [4], our MGDM model demonstrates superiority over existing supervised methods.
>
> **Weakness 2:** We recognize your concerns about the conjugate symmetry in k-space when applying Fourier Transform to its real-valued images like BraTS, which may affect the effective undersampling factor, potentially halving the apparent ACR. I addressed this issue in my work and included a citation to the paper you have recommended, which discusses the biases that can arise from the misuse of public data. While I continue to utilize the BraTS dataset for its merits, I ensure to mention this potential pitfall for clarity and accuracy.
>
> Quoted from our limitation part: “Additionally, the BraTS dataset, employed both in our study and by the baseline methods, has been indicated in a recent paper (Shimron et al., 2022) to have an overestimated undersampling factor, which arises from the conjugate symmetry of k-space inherent in real-valued images.”
>
> Additional results for the FastMRI dataset, including undersampling patterns and detailed descriptions, are provided in the main paper and Appendix.
>
> **Weakness 3:** We first want to acknowledge that other baseline methods have also reported the capability to reconstruct MRI BraTs datasets under x24 acceleration factor. For example, in Song’s paper [4], they reported a similar PSNR of 29.42; ours is slightly higher, with a PSNR of 30.04. We believe that learning informative priors can significantly aid in signal recovery. The training and testing datasets in Brats come from the same distribution, so we expect to achieve good results and effectively recover brain tumors. However, if tested on a dataset outside this distribution, our method may not perform as well, which would require additional adaptations. Our reasoning is further supported by our robust data consistency step.
>
> **Minor Problem 1:** We apologize for the oversight and thank you for bringing it to my attention. Now, all figures, including Figure 5, are provided in high resolution without compression artifacts in the revised manuscript. Additionally, the orientation of the knee images is corrected to the standard presentation.
>
> **Minor Problem 2:** The citation for ESPIRiT is now corrected, and a thorough review of all other references is conducted to ensure their accuracy. Thank you for pointing out this discrepancy.
>
> **Question 1:** Please refer to our response to Weakness 3.
>
> **Question 2:** We have now added Figure 6 to show how the reconstruction results look like with different ablations.
>
> [1] Hyungjin Chung, Jeongsol Kim, Michael T Mccann, Marc L Klasky, and Jong Chul Ye. Diffusion posterior sampling for general noisy inverse problems. arXiv preprint arXiv:2209.14687, 2022a.
>
> [2] Yang Song, Liyue Shen, Lei Xing, and Stefano Ermon. Solving inverse problems in medical imaging with score-based generative models. arXiv preprint arXiv:2111.08005, 2021.
>
> [3] Yinhuai Wang, Jiwen Yu, and Jian Zhang. Zero-shot image restoration using denoising diffusion null-space model. arXiv preprint arXiv:2212.00490, 2022.
>
> [4] Bahjat Kawar, Michael Elad, Stefano Ermon, and Jiaming Song. Denoising diffusion restoration models. Advances in Neural Information Processing Systems, 35:23593–23606, 2022.

---

### Official Review · Reviewer_TLju · 2023-10-27

**Soundness:** 2 fair
**Presentation:** 2 fair
**Contribution:** 2 fair
**Rating:** 3
**Confidence:** 3

**Summary:**

This paper is tasked with using diffusion models to solve medical imaging inverse problems.
It proposes a new method based on an introduced optimization problem to derive the sampling of images from a diffusion model conditioned on measurements (typically the k-space or sinogram).
A series of experiments is then presented to showcase the effectiveness of the method

**Strengths:**

- the experiments are diverse and a ablation study is presented to get more insights
- the problem tackled is interesting and in a very thriving area of research

**Weaknesses:**

- **Presentation**: the presentation needs to be reworked: for example one page is dedicated to introducing diffusion models, which I think is unnecessary; a few lines would suffice to introduce relevant notations and point to relevant references. There are a lot of typos which can be checked using grammarly or LTex (https://valentjn.github.io/ltex/vscode-ltex/installation-usage-vscode-ltex.html) or weird formulation (why zero-shot?).
In addition, there are too many confusing notations, and it's difficult to piece how they interact together even with the figure or the algorithm (which would benefit from comments): for example when the ablation study is conducted, I don't what equations the different labels mentioned refer to.
- **Method**: the presentation of the method is very handwavy : there is no derivation of why this sampling is supposed to work even with very strong assumptions. The only "theoretical" grounding is proposition 3.1 which to me amounts a bit to mathiness given its simplicity. Another example is the introduction of the discrepancy gradient which is not discussed.
- **Results**: given how good the results are, it's important to question why the improvement is so big. If I focus on knee MRI reconstruction, there is a +4dB improvement in PSNR: this is absolutely huge but it isn't discussed. In particular, it should be viewed in comparison with fully supervised methods like the ones presented in the 2019 fastmri challenge which are nowhere near the performance reported here.
- **Code**: while some code is provided, the README has not been updated from the Wang et al. repo, which makes it difficult to know how to look at the relevant code, i.e. where the new sampling method is introduced.


Nitpicks:
The paper is 10 page-long rather than 9, but it's just due to a figure that slipped into the 10th page.

**Questions:**

- How can the new sampling be derived using a bayesian formulation?
- How can you explain the gap between this work and previous works?
- What is (pr) i.e. proximal optimization and refinement?

---

> ### Author Response · Authors · 2023-11-22
> **Official Response of Submission7428 (Part 1)**
>
> We thank the reviewer for their constructive comments. We highly invite the Reviewer to read our paper again, we have significantly improved our representation of the work, including typos, notations, concepts, and formulations.
>
> Below are our point-by-point responses to Weaknesses and Questions:
>
> **Weakness 1 (Presentation):** We have now reworked our presentation of the paper with major changes highlighted in blue.
>
> * **Diffusion Models:** We had initially aimed to make the paper as comprehensive as possible. Following your suggestion, we have now updated that section by removing the continuous part and rendering the discrete section more concise.
>
> * **Typos:** Thank you for your feedback. We have now thoroughly checked grammar and polished our writing. If you notice any further discrepancies, please know they will be addressed in our continuous efforts to improve.
>
> * **Zero-Shot:** Zero-shot learning is one of the several learning paradigms aimed at out-of-distribution generalization, where the algorithm is trained to categorize objects or concepts that it has not been exposed to during training. In this paradigm, the set of classes during training, denoted as $\mathcal{Y}^{train}$, is distinct and non-overlapping with the set of classes during testing, denoted as $\mathcal{Y}^{test}$. The challenge is for the model to generalize from the training classes to the test classes, a significant leap as it must make accurate predictions for classes without having any direct prior examples to learn from. For inverse problems, where the distribution of measurements $\mathcal{Y}$ can change based on the undersampling pattern, the term 'zero-shot solver' is sometimes applied. It also follows the precedent set by prior works that have employed this terminology for plug-and-play approaches, such as DDNM and SSD [1]. This usage is intended to reflect the solver’s ability to adapt to different undersampling patterns without retraining, similar to how zero-shot learning generalizes to new classes without prior exposure.
>
> * **Confusing notation:** Regarding your concerns on the confusing notations, we've reworked our notations in all mathematical equations and algorithms to make them consistent and easy to follow. Also, we would happily address any, if you could kindly point us to specific places where confusion might arise.
>
> **Weakness 2 (Method):**
> * **Derivation:** We understand that although we did not explicitly state that our approach is derived from Bayesian principles, some terminology used in the previous introductory presentation could have led to confusion that our approach is based on Bayesian inference.  Therefore, we have decided to revise the presentation accordingly. The modified texts are now highlighted in blue in the revised manuscript and as following:
>
>   > "To mitigate the ill-posedness, it is essential to incorporate an additional assumption based on _prior_ knowledge to constrain the space of possible solutions. In this manner, the inverse problem then can be addressed by optimizing or sampling a function that integrates this prior or regularization term with a data consistency or likelihood term {ongie2020deep}. A prevalent approach for prior imposition is to employ pre-trained deep generative models {bora2017compressed, jalal2021robust}."
>
> * **Theory:** We respectfully disagree with the reviewer’s sentiment of the theoretical foundation in our work and we find their characterization of our results to be, rather, unfair. We have designed an algorithm that is to a great extend theoretically motivated. Proposition 3.1, as the name suggest, only small piece of our work. Another important part of our work is the practical side, which clearly shows that our findings and analysis are on the right track.
>
> * **Discrepancy Gradient step:**
> Regarding, discrepancy gradient step, we added some information in the paper and our reasoning.
> Discrepancy gradient guides $\mathbf{x}_{t-1}$ towards an equilibrium between two values x_hat_0|t and x_tilde_0|t. This step serves to add additional corrections based on the specific characteristics of the problem at hand, such as incorporating more detailed information from the measurement $\mathbf{y}$ or adjusting for errors of approximations made in previous steps. This adjustment can mitigate the effects of any errors introduced in earlier steps, thus ensuring a more consistent and robust convergence toward an optimal solution. Empirical evidence suggests that this step can moderately enhance the reconstruction results. We have now explained this clearly in the text.
>
> [1] Gongye Liu et al. Accelerating Diffusion Models for Inverse Problems through Shortcut Sampling. 2023.

---

> > ### Author Response · Authors · 2023-11-22
> > **Official Response of Submission7428 (Part 2)**
> >
> > **Weakness 3 (Results):**
> >
> > We appreciate your comments about our results. After thorough checking, we noticed that we had reported the unifrom1D instead of the Gaussain1D from the Score-MRI paper. We apologize for this insight and have now corrected their values in Table 2.
> >
> > | Method                                   | 4x ACR PSNR↑     | 4x ACR SSIM↑    | 8x ACR PSNR↑     | 8x ACR SSIM↑    |
> > |------------------------------------------|------------------|-----------------|------------------|-----------------|
> > | DPS (Chung et al., 2022a)                | 22.41±3.33       | 0.650±0.080     | 21.87±2.91       | 0.607±0.076     |
> > | DDNM (Wang et al., 2022)                 | 35.87±2.68       | 0.873±0.065     | 34.04±2.70       | 0.847±0.071     |
> > | Score-MRI (Chung & Ye, 2022)             | 33.96±1.27       | 0.858±0.028     | 30.82±1.37       | 0.762±0.034     |
> > | MGDM (ours)                              | **36.94±2.70**   | **0.888±0.062** | **34.98±2.66**   | **0.856±0.070** |
> > |------------------------------------------|------------------|-----------------|------------------|-----------------|
> >
> > Additionally, in Table 4 of the Appendix, we present supervised results. Consistent with other zero-shot inverse problem solvers, such as those discussed in Chung et al. (2022), Wang et al. (2022), and Kawar et al. (2022), our MGDM model demonstrates superiority over existing supervised methods.
> >
> > **Weakness 3 (Code):**
> > We would like to clarify that the code supporting the findings of our study is based on three GitHub repositories, each corresponding to a dataset: DDNM by Wang et al., Score-Med by Song et al., and Score-MRI by Chung et al. We need time to enhance the documentation and improve the code's readability. We recognize the importance of reproducibility in scientific research and plan to make the code publicly available once these improvements are made. In the meantime, I have added a simple README file that explains how to run the code.
> >
> > in our paper, we do not claim to have derived our algorithm directly from Bayesian inference. While we initially discussed Bayesian inference, this aspect has since been refined and is not the basis for our algorithm's derivation.
> >
> > To address your concerns regarding the gap, we have incorporated additional lines in our revised manuscript.
> >
> > **Question 1:** In our paper, we do not claim to have derived our algorithm from Bayesian inference. While we initially discussed Bayesian inference, this aspect has since been refined and is not the basis for our algorithm's derivation.
> >
> > **Question 2:** To address your concerns regarding the gap, we have incorporated additional lines in our revised manuscript.
> >   > A central challenge in zero-shot inverse problem solving is how to guide an unconditional prediction to conform to the measurement information. Existing methods rely on deficient projection or inefficient posterior score approximation guidance, which often leads to suboptimal results. In this paper, we propose a Meta-Guided Diffusion Model (MGDM) that tackles this challenge through a _bi-level_ guidance strategy, where the _outer level_ solves a proximal optimization problem to impose measurement consistency and the _inner level_ approximates the measurement-conditioned posterior mean as the initial prediction.
> >
> > **Question 3:** We have clarified these points in the text for better understanding. The proximal optimization process is outlined as Step 10 in Algorithm 2. Additionally, the 'discrepancy gradient step' is synonymous with what we refer to as the 'refinement step’.
> >
> > 1. The exclusion of **proximal optimization** (step **9**) and **refinement** (step **11**) termed `no-pr`.
> > 2. The omission of **initial prediction** (step **8**) and **refinement** (step **11**) designated as `no-ir`.
> > 3. The absence of **initial prediction** alone (step **8**) noted as `no-i`.
> > 4. The removal of **refinement** alone (step **11**) referred to as `no-r`.

---

> > > ### Comment · Reviewer_TLju · 2023-11-22
> > >
> > > I would like to thank the authors for engaging in the discussion respectfully.
> > >
> > > I think the 2 main weaknesses are not satisfactorily addressed:
> > > - the derivation of the algorithm is still handwavy. Typically, for eq (13) it says in the revised paper:
> > > > . In practice, we assume that $p_t(y|x_t) \approx \mathcal{N}(y; Ax_{0|t},\sigma^2_t, I)$, and then an approximation of the
> > > expectation in Eq. (12) is : $\tilde{x}_{0|t} \approx \frac{1}{\sqrt{\bar{\alpha}_t}} [x_t - \sqrt{1 - \bar{\alpha}_t} \epsilon_\theta (x_t, t) \zeta \nabla_{x_t} \|y - Ax_{0|t}\|_2^2]$
> > >
> > > Why do make this assumption? What exactly is this assumption : $\approx$ could mean several different things and it's not clear what here (and could be different in the 2 equations)? What are the assumptions needed for this result to hold? Are they relevant/realistic?
> > > Similarly when justifying the use of the discrepancy gradient, it says in the revised paper:
> > > > This adjustment can mitigate the effects of any errors introduced in earlier steps, thus ensuring a more consistent and robust convergence toward an optimal solution.
> > >
> > > But there is no characterization of said errors, and how in principle this discrepancy gradient could mitigate that formally and under which assumptions.
> > >
> > > To me these are all huge problems, and this comes in addition to what I again think is a "mathy" proposition 3.1, which I understand is not the core of the paper. Still the theoretical foundations for such a work claiming extraordinary empirical results need to be very clear.
> > >
> > > - The numbers reported by Chung and Ye are the ones reported in Table 4. Those numbers are overly pessimistic when it comes to comparing against fully-supervised methods. Indeed they specify in this paper: "We use the official implementation5
> > > , with 4 recurrent blocks and default parameters [...] For all the deep learning comparison studies, we train the network with Gaussian 1D random sampling masks.". This means they used parameters that were not tailored to this specific mask, which is unrealistic (it's much better to use a cartesian mask to be closer to clinical practice). The fact that retraining happened makes also the reported numbers much less valid. The ideal experiment would be to test these diffusion methods against fully-supervised methods pre-trained using their actual masking/undersampling methods.
> > > Finally, the numbers reported in Table 4 are not those of DuDorNet as mentionned, but those of E2E-VarNet which is multi-coil and not single-coil as what is done in this paper.
> > > I think it would generally be very surprising if for a given problem setting (i.e. fixed modality and undersampling pattern), an unsupervised method using the same amount of data as a supervised method performed better. Here the fully-supervised method can adapt to the specific aliasing patterns in the data.
> > > Still, even not considering fully-supervised methods, the gap between the introduced method and the previous state of the art is so huge it needs a better in-depth explanation as to the failure modes of the previous methods and why this new one succeeds.
> > >
> > >
> > > As another point, code is still not ready for review. I couldn't find a clear explanation of how to reproduce the results.

---

> > > > ### Author Response · Authors · 2023-11-23
> > > > **Response to Equation derivations**
> > > >
> > > > Dear Reviewer,
> > > >
> > > > We would like to address your questions and provide an update on our manuscript:
> > > >
> > > > **Equations Derivation:** We have now provided thorough derivations from Eq. 12 to Eq. 13 in the Appendix 4.2. Due to the Markdown limitation, we cannot properly display the equations here, can we please invite the Reviewer to read our new PDF submission. To address your question about the assumptions for deriving Eq 13, they are now highlighted in blue (above the equation) and further explained in Appendix 4.2. These assumptions are based on the papers by DPS (Chung et al., 2022a) [1] and Ravula et al. (2023) [2]. We kindly invite you to review the new PDF submission for a more detailed explanation.
> > > >
> > > > **Discreprency Gradient**: We acknowledge your observation regarding the lack of a mathematical proof for the 'discrepancy gradient' step. Upon reflection, we concur with your assessment. While this step has shown empirical benefits, we acknowledge that it lacks formal mathematical proof and remains primarily an empirical finding. Our high-level interpretation of this step is that it aids in improving the accuracy of the approximated measurement-conditioned posterior mean (x_hat_0t) and reduces the necessity of the projection step (9). However, given that the improvement it brings is moderate, we have decided to restrict our descriptions to 'empirical findings.' We recognize the importance of further research in this area and intend to explore it in the future.
> > > >
> > > > [1] Hyungjin Chung, Jeongsol Kim, Michael T Mccann, Marc L Klasky, and Jong Chul Ye. Diffusion posterior sampling for general noisy inverse problems. arXiv preprint arXiv:2209.14687, 2022a.
> > > > [2] Sriram Ravula, Brett Levac, Ajil Jalal, Jonathan I Tamir, and Alexandros G Dimakis. Optimizing sampling patterns for compressed sensing mri with diffusion generative models. arXiv preprint arXiv:2306.03284, 2023.

---

> > > > > ### Comment · Reviewer_TLju · 2023-11-23
> > > > >
> > > > > **Equations derivation**: the equation derivation in Appendix 4.2. still uses $\approx$ without specifying when the approximation becomes true, i.e. under which assumptions, and whether these assumptions are close to something reasonable.
> > > > > I want to highlight that it is not the only problem when it comes to mathematical derivations in the paper and it was more of an example. I think the whole theoretical section needs to be reworked in that sense.
> > > > >
> > > > > **Discrepancy gradient**: I think it's fair to have empirical findings and they should just be described as such.
> > > > > However, I see that most of the discussion at the moment happens in the appendix, which is also a problem for the paper: limitations and discussion should be put forth in the main text.

---

> > > > ### Author Response · Authors · 2023-11-23
> > > > **Response to code readability**
> > > >
> > > > We've provided datasets, pre-trained models, and code in the supplementary file via the Google Drive link due to the space constraint of 100 MB on OpenReview. Please access the code and steps to reproduce results using the provided link. Also, refer to the README file for setting up the environment and instructions to run the code.

---

> > > > > ### Comment · Reviewer_TLju · 2023-11-23
> > > > >
> > > > > Regarding code, I think it's normal to have a size limit in the ICLR submission.
> > > > > For the next round of revision or for the next paper, I invite the authors to follow the guidelines written here: https://nips.cc/Conferences/2020/PaperInformation/CodeSubmissionPolicy
> > > > >
> > > > > Basically, source code should be zipped along with small datasets, but large datasets should just be downloaded from the internet.

---

> > > > > > ### Comment · Reviewer_TLju · 2023-11-23
> > > > > >
> > > > > > I tried running the code just now, and I have too many errors to fix while just setting up the environment: I think the instructions for the setup have bugs (for example there are references to local files in the `requirements.txt`, some packages like the `mkl` ones are not available through `pip`).

---

> > > > > > > ### Author Response · Authors · 2023-11-23
> > > > > > > **Running Code**
> > > > > > >
> > > > > > > Could you please let us know if you were able to run the code. If not, could you please install DDNM dependencies first and then if you need other dependencies (probably one or two more) install them.

---

> ### Author Response · Authors · 2023-11-23
> **Response on comparison with supervised approach**
>
> Dear Reviewer,
>
> We appreciate your feedback and would like to first provide some response about your concern on comparison with supervised approach:
>
> **Sampling Mask Clarification:** We would like to clarify that the term 'Gaussian 1D random sampling mask' actually refers to a 1D Cartesian mask. We want to make sure the Reviewer has not confused this with 'Gaussian 2D or Possoin Disk mask', which are commonly used for accelerating 3D MRI. The current Gaussian 1D random sampling mask features a full-acquisition in the readout dimension and pseudo-randomly skipped sampling steps in the phase encoding dimension. Importantly, this approach aligns with clinical practice, as many 2D MRI acquisitions employ similar 1D randomly-distributed sampling masks for compressed-sensing-based image acceleration. It's worth noting that this 'random' 1D Cartesian sampling mask pattern is applied deterministically during the pulse sequence design. We want to highlight that our last author possesses a substantial background in MRI, encompassing pulse sequence design and image reconstruction, ensuring the clinical relevance of our work.
>
> **Comparison with Fully-Supervised Methods:** We acknowledge your point about the potential disadvantage of the 'random' sampling mask approach in the context of training the supervised methods. Since training data containing a mix of different sampling mask effects can be challenging for supervised approaches. Although we believe that the Gaussian 1D distribution underpinning these masks may mitigate the mixture effect to some extent, as these masks are very similar to some degree. However, conducting a comprehensive comparison against fully-supervised methods falls beyond the scope of our current study. Implementing various state-of-the-art supervised methods within our revision timeline is a complex undertaking. Hence, we opted to refer to previous publications and quote relevant numbers. We have made efforts to adjust and clarify these comparisons in Appendix Table 4 to highlight the disadvantages that the 'Gaussian 1D random sampling mask' presents to supervised learning approaches. If you strongly believe that the supervised performance reported in the Chung and Ye paper is biased, we are open to removing Table 4 from our Appendix. After reevaluating the tables in the Chung and Ye paper, we can confirm that we are reporting the DuDorNet numbers, not those of E2E-VarNet. We kindly request your review of this clarification.
>
> As mentioned in our response to 'Weakness 3 (Results),' we have rectified the numbers we previously misreported. The corrected results indicate an improvement of approximately 1dB over the state-of-the-art, which we believe represents a reasonable advancement. We apologize for any confusion arising from our previous use of a different sampling mask.
>
> Thank you for your continued feedback and consideration. We are currently working on writing down the Equations deriving E. 13 from Eq 12, which will be posted soon.

---

> > ### Comment · Reviewer_TLju · 2023-11-23
> >
> > **Comparison with fully-supervised**: I was wrong on the sampling mask indeed. I was also wrong on the reported results and for this I want to apologize especially since these mistakes happen so late in the discussion period.
> >
> > I think I was confused when it comes to the sampling mask by both the denomination and the fact that Chung and Ye retrained the network: if the same sampling mask is used, then retraining the network shouldn't be necessary, except for different data, in which case the argument for bad hyperparameter setting holds.
> > I agree that conducting a full comparison is beyond the scope of this study, nonetheless, it's worth noting the following discrepancy: how can unsupervised models (in the sense that they don't know the undersampling mechanism) with access to the same amount of data and compute, outperform supervised models (in the sense that they are aware of the undersampling mechanism)? This to me indicates rather a problem in evaluation than anything else.
> >
> > **Results update**: the fact that the results changed from a margin of 4dB to now a margin of 1dB is a red flag: this means that there are problems in the evaluation, and the fact that code was so difficult to put together in a state where it's reviewable is another evidence of that. I think because the empirical results have so substantially changed, this paper needs in any case to undergo another round of review. I also think it's important to highlight this change in the results in the revision in blue, like what was done for the rest of the paper.

---

> > > ### Author Response · Authors · 2023-11-23
> > > **Office comment**
> > >
> > > Thank you for your feedback. We appreciate the Reviewer's insightful comments regarding our discussion of the Chuang and Ye paper.
> > >
> > > **Fully-Supervised vs. Unsupervised Training:** We acknowledge the Reviewer's skepticism about unsupervised training outperforming supervised training as reported in the Chuang and Ye paper. Our interpretation is that this phenomenon may arise from the use of 1D random Gaussian sampling masks. Each 2D k-space training sample is undersampled with a uniquely varying mask, leading to a diverse training dataset. This diversity can impede the performance of supervised learning due to the inconsistency in undersampling patterns. It is akin to combining training data with varying acceleration factors, such as x4 and x8, which poses a significant challenge for supervised methods. Conversely, unsupervised models like diffusion models are inherently more adaptable to different sampling masks. Alternatively, it is also possible that the reported results in the paper might be incorrect, or the supervised networks were not optimally tuned.
> > >
> > > As mentioned in our previous response, if the Reviewer believes that these results are unreliable, we are open to removing them from the Appendix. We value your guidance on this matter and look forward to your suggestions.
> > >
> > > **Results Update and Correction:** We sincerely apologize for the error in reporting values from the Chung and Ye paper in our initial submission. We recognize that the paper contains numerous tables with varied naming conventions and a multitude of methods, which led to our inadvertent referencing of data associated with a different sampling mask. This oversight was unintentional, and we regret any confusion it may have caused.
> > >
> > > We would like to clarify that our results for MGDM (our proposed method) remain unchanged, consistent with our original findings derived from our experiments and coding. These results are a direct product of our research and have not been altered in the revision process.
> > >
> > > Furthermore, we are actively working to streamline the code execution environment. Our aim is to ensure that the Reviewer can personally execute the codes and verify the results we have reported concerning MGDM. We value transparency and reproducibility in our research, and we are committed to providing all necessary resources for an independent evaluation of our work. Your understanding and guidance in this matter are greatly appreciated.

---

> > > > ### Comment · Reviewer_TLju · 2023-11-23
> > > >
> > > > I think my main point at this stage in the discussion is the following: there were too many problems with evaluation in the original version, the results have therefore changed (i.e. the ones reported for Wang et al. in Table 1, that had not been previously reported in the literature if I understand correctly) and therefore I think the paper needs to go through another round of reviews.
> > > >
> > > > Furthermore, the theoretical part needs to be rewritten to adhere to the standards of the community in terms of rigor for the derivations.

---

> > > > > ### Author Response · Authors · 2023-11-23
> > > > > **Office response**
> > > > >
> > > > > We extend our sincere thanks to the Reviewer for their valuable and constructive feedback, and for engaging in a fruitful discussion with us.
> > > > >
> > > > > In relation to the modifications in the DDNM results, we acknowledge that there was a mix-up in reporting between '1D uniform sampling mask' and '1D Gaussian sampling mask'. In our earlier submission, the results were inadvertently reported using the 'uniform' sampling mask, whereas it should have been the 'Gaussian' sampling mask. This discrepancy was promptly addressed and corrected following the Reviewer's observations in the initial review.
> > > > >
> > > > > We would like to emphasize that diffusion models, including our model and DDNM, generally exhibit enhanced performance with the 'Gaussian' sampling mask. This is attributed to the greater significance of the central part of the k-space measurements in the image reconstruction process, particularly for guiding the model. The outer regions of k-space, which are more relevant for high-frequency image details, play a less critical role in this context. We regret the confusion caused by our initial mixed reporting of the results from different sampling masks, and we are committed to ensuring the accuracy and clarity of our data in all future communications.

---

> ### Author Response · Authors · 2023-11-23
> **Official response**
>
> We are grateful for the Reviewer's constructive feedback regarding the presentation and clarity of our equations. We acknowledge that some of the derivation steps may not have been immediately clear, especially to readers who are not intimately familiar with the foundational works of DPS (Chung et al., 2022a) [1] and Ravula et al. (2023) [2].
>
> In response to your critique, we have refined these derivations to enhance their readability and accessibility. This includes incorporating relevant assumptions directly following each equation in the Appendix 4.2, rather than grouping them together in the main document prior to Equation 13. We believe this approach has provided a clearer and more intuitive understanding of our methodology.
>
> The only assumption made here is measurement noise is Gaussian, which is a standard assumption in most MRI simulation studies. All other \simeq are approximations derived in DPS (Chung et al., 2022a) [1] and Ravula et al. (2023) [2]. These are now clearly explained together with the Equations both in the main document and in Appendix.

---

### Official Review · Reviewer_Bg7d · 2023-10-31

**Soundness:** 3 good
**Presentation:** 3 good
**Contribution:** 2 fair
**Rating:** 6
**Confidence:** 3

**Summary:**

The paper proposes a new method for inverse problem in medical imaging. The goal is to apply Diffusion Models in medical imaging to produce high-quality images using incomplete and noisy measurements, aiming to reduce costs and risks to patients. To this end, a model named Meta-Guided Diffusion Model (MGDM) is introduced to address the challenge of guiding unconditional predictions to align with measurement information through a bi-level guidance strategy (an outer level and an inner level ). The outer level optimizes for measurement consistency, while the inner level approximates the measurement-conditioned posterior mean as the initial prediction. Empirical results on medical datasets in MRI and CT demonstrate that MGDM outperforms existing methods by generating high-fidelity medical images that closely match measurements and reduce the occurrence of hallucinatory images.

**Strengths:**

-  This paper provides an effective strategy for addressing medical imaging inverse problems in a zero-shot setting.
-  Empirical results show a clear improvement, consistently overcoming the state-of-the-art benchmarks, and exhibiting robustness across diverse acceleration rates, projection counts, and anatomical variation.

**Weaknesses:**

- Seems the method is a combination of DPS (Chung et al., 2022a) and DDNM (Wang et al., 2022), which somehow limits the contribution of this paper.
- The 3D volumes are divided into 2D slices. How can the method ensure the consistency of the volume from other views, like sigital and coronal?

**Questions:**

- What do you mean by 'a novel class of "fully" probabilistic Deep Learning Models (DLMs)'?
- Seems in section 2.1 that there is no need to include both Discrete-time formulation and Continuous-time formulation, just introduce the one related to this paper.
- the proximity term penalizes deviations from the initial estimate, which seems like the data consistency in MRI acceleration?
- Is the pre-trained Guided diffusion model (from natural images) used here or the model is trained from scratch?
- Why did not test on the brain data from fastMRI? Can we share the same model for BraTS and brain from fastMRI?
- I am not quite sure what the mentioned 'zero-shot setting' is? For MRI accereration, we do have the fully sampled k-sapce data as the reference to train the neural network. What is the 'zero-shot setting' here?
- Can we share the model for different acceleration rates?

---

> ### Author Response · Authors · 2023-11-22
> **Official Response of Submission7428 (Part 1)**
>
> We thank the reviewer for their constructive comments. Below are our point-by-point responses to Weaknesses and Questions:
>
> **Weakness 1:**
>
> We respectfully disagree with the comment that our proposed method “is a combination of DPS and DDNM”. DDNM is given by the expression $$
> \hat{\mathbf{x}} = {\mathcal{A}^{\dagger}\mathbf{y}}+{(\mathbf{I}-\mathcal{A}^{\dagger}\mathcal{A})}{\bar{\mathbf{x}}}
> $$, representing a direct analytical approach, which utilizes a straightforward formula involving matrix operations. The solution is a constructive combination of the least-squares fit and the null space component, aiming to be as faithful to the data measurement as possible while respecting the initial prediction. This is a deterministic and direct method, which clearly follows the path dictated by the data and initial conditions.
>
> On the other hand, MGDM has a closed-form solution derived from balancing two terms in the objective function.
> $$
> \hat{\mathbf{x}}_{t} = \underset{\mathbf{x}}{\mathrm{argmin}}  \frac{1}{2} ||\mathbf{y} -\mathcal{A}\mathbf{x}||_2^2 + \frac{\lambda}{2} ||\mathbf{x}-\mathbf{x}_0t||_2^2
> $$
>
>  MGDM comes from the closed-form of the optimization problem, reflecting an optimization-based approach. It aims to find the best route that respects closeness to the path. The closed-form solution embodies a compromise that includes fitting the model to the data and adhering to a prior reference x0|t, which is introduced through the regularization term λ. The solution represents an equilibrium between minimizing the least-squares error and the regularization term, even when expressed in closed form. This method represents an iterative approach to balance, between fitting the data and incorporating prior knowledge, reflecting a philosophical stance even in the presence of a closed-form solution.
>
> Secondly, our MGDM method differs from the DPS method's purpose and application of gradient-based updates. While both methods involve gradient-based updates, they serve distinct roles. In Algorithm 2 (step 8) of our MGDM, the gradient-descent update is used to refine our clean image initial prediction with the measurement y as a conditioning factor (i.e., approximated measurement-conditioned posterior mean). This contrasts DPS, where gradient-based updates are applied to the noisy image after each reverse sampling process.
>
> In response to your valuable feedback, we have revised our text to ensure that these distinctions between our method, DDNM, and DPS are clearer.
>
>
> **Weakness 2:** We appreciate the Reviewer's inquiry and understand the confusion regarding the differences in acquisition modes, particularly from a non-medical-imaging perspective. For instance, in MRI, acquiring only a few 2D slices with anatomical gaps between them is common practice to save scan time. In such cases, stacking them into a continuous 3D volume is not feasible.
>
> In our approach, we focus on reconstructing 2D images from individual 2D slices obtained from multi-slice 2D acquisitions. Each slice is treated as an independent entity, and our primary goal is to reconstruct the best possible 2D image from the raw 2D slice measurements. This is consistent with other SOTA methods we are comparing against in our paper.
>
> However, suppose one decides to acquire multi-slice 2D images covering the 3D slab without gaps between the slices, as the Reviewer suspected. In that case, potential discontinuities may indeed arise when stacking these 2D images to form a 3D volume in other views, such as coronal or sagittal. It's important to note that these discontinuities are inherent limitations of multi-slice 2D acquisitions and are not specific to our reconstruction methods.
>
> To address this limitation and ensure smoother transitions in the 3D volume, one viable solution is presented in a recent paper [1]. This approach introduces Total Variation regularization in the Z-dimension, effectively enforcing a smooth transition between slices in the 3D volume.
>
> We highly value this valuable feedback and have revised our text to emphasize the distinction between multi-slice 2D acquisitions and full-slab 3D acquisitions. Additionally, we have referenced the mentioned paper to provide readers with further insights into addressing the challenges associated with 3D reconstruction from 2D diffusion models.
>
> [1] Lee, Suhyeon, et al. "Improving 3D Imaging with Pre-Trained Perpendicular 2D Diffusion Models." arXiv preprint arXiv:2303.08440 (2023).

---

> > ### Author Response · Authors · 2023-11-22
> > **Official Response of Submission7428 (Part 2)**
> >
> > **Question 1**: By fully probabilistic deep models we in fact meant deep generative models. Generative models can indeed be considered a subset of fully probabilistic deep learning models. Recall that fully probabilistic models capture the entire probability distribution of the dataset.  They can be used for a wide range of tasks, including but not limited to prediction, regression, and generation.  On the other hand, generative models are specifically designed to sample and generate new data points that are like the ones they were trained on. To do this, they must learn the probability distribution of the dataset. Therefore, they inherently deal with the probabilistic nature of the data. In essence, generative models are one application of fully probabilistic deep learning models, with the specific task of data generation. The term "fully" in the context of fully probabilistic deep learning models is meant to convey completeness in terms of probabilistic modeling.
> >
> > However, to avoid confusion, we have now replaced "fully probablistic deep learning models" with "deep generative models".
> >
> > **Question 2**: We had initially aimed to make the paper as comprehensive as possible. Following your suggestion, we have now updated that section by removing the continuous part and rendering the discrete section more concise.
> >
> > **Question 3**: As explained in the introduction and method sections, the proximity term in Eq [10] ensures the solution remains close to its initial prediction estimated from the denoiser pre-trained by diffusion models. This is not about the data consistency in MRI. In fact, the fidelity term in Eq [10] enforces that the reconstructed image is consistent with the acquired measurements in the transformed domains. We have now made this clear in the text.
> >
> > **Question 4**: The models are trained by ourselves from scratch on the ADM architecture using the public MRI and CT datasets, not pre-trained checkpoints from natural images. We have now made this clear in the text by highlighting in blue color.
> >
> > **Question 5**: The choice of dataset for model validation is crucial for demonstrating the model's generalizability and robustness. In our current study, we focused on the BraTS dataset due to its diverse image contrasts and tumor pathologies, as also adopted by a baseline method we are comparing against [1].
> >
> > To address your point on model sharing between BraTS and fastMRI, it is possible to apply the same model to both datasets. However, adaptations for out-of-distribution (ODD) are necessary to account for the different characteristics and objectives of each dataset. We acknowledge the importance of evaluating our model on multiple datasets to establish its generalizability, and we did this on different modalities and anatomical choices. We also are considering fastMRI data in future work to validate our model's performance for ODD analysis.
> >
> > **Question 6**: We want to make sure this is clear to the Reviewer that different from existing supervised methods using the fully sampled k-space as the reference (label) for training, our approach does not involve any paired data (undersampled and fully-sampled k-space) into our unsupervised training process. The undersampled k-space is only used for sampling/reconstructing the corresponding MRI image during inference time.
> >
> > Zero-shot learning is one of the several learning paradigms aimed at out-of-distribution generalization, where the algorithm is trained to categorize objects or concepts that it has not been exposed to during training. In this paradigm, the set of classes during training, denoted as $\mathcal{Y}^{train}$, is distinct and non-overlapping with the set of classes during testing, denoted as $\mathcal{Y}^{test}$. The challenge is for the model to generalize from the training classes to the test classes, a significant leap as it must make accurate predictions for classes without having any direct prior examples to learn from. For inverse problems, where the distribution of measurements $\mathcal{Y}$ can change based on the undersampling pattern, the term 'zero-shot solver' is sometimes applied. It also follows the precedent set by prior works that have employed this terminology for plug-and-play approaches, such as DDNM and SSD [2]. This usage is intended to reflect the solver’s ability to adapt to different undersampling patterns without retraining, similar to how zero-shot learning generalizes to new classes without prior exposure.
> >
> > **Question 7**: Yes, as explained in response to question 6, the nature of zero-shot image reconstruction allows us to adapt the model for various acceleration rates. All the results reported for each test dataset on different acceleration factors share the same model for that dataset.
> >
> > [1] Yang Song et al. Solving inverse problems in medical imaging with score-based generative models. 2021.
> > [2] Gongye Liu et al. Accelerating Diffusion Models for Inverse Problems through Shortcut Sampling. 2023.

---

> > ### Comment · Reviewer_Bg7d · 2023-11-22
> >
> > Thanks for the explanation and response. I would like to keep my score, also based on comments from other reviewers.

---

### Official Review · Reviewer_sAeb · 2023-11-01

**Soundness:** 3 good
**Presentation:** 3 good
**Contribution:** 3 good
**Rating:** 6
**Confidence:** 4

**Summary:**

Sparse-data reconstruction in CT and MR imaging is modeled as an ill-posed linear inverse problems subject to noise.
Similar to the DDNM (Denosing Diffusion Null-space models) algorithm for MRI reconstruction, the authors use diffusion denoising,
but replace the backprojections in DDNM with a bi-level MGDM approach relying upon a regularized outer objective and an inner expectation approximation.
Simulated comparisons with competing approaches demonstrate the efficacy of the proposed approach.

**Strengths:**

1. The authors demonstrate improvements in terms of pSNR and SSIM (an image quality metric) over competing algorithms for both fastMRI datasets via simulation and for CT simulations using real LIDC CT reconstructions as digital phantoms.

2. The ablation study in Sec. 4.4 and Table 3 clearly demonstrates that the regularized proximal optimization plays the most substantial role in the proposed MGDM method.

**Weaknesses:**

1. Two extra parameters $\zeta$ and $\rho$ are introduced in the proposed Algorithm 2 (MGDM sampling) in comparison to Algorithm 1 (DDNM sampling).
It is clear not how much of the improvements over DDNM were obtained by painstakingly tuning these two new parameters.

2.
While a slice-wise 2D imaging simulation is appropriate for demonstrating the practical efficacy for MR reconstruction and
the MRI simulation, as in the fastMRI paper, appears to be realistic, the 1989 ESPIRIT reference for estimate the parallel multi-coil sensitivities appears to be incorrect.
The cited 1989 paper doesn't consider any special considerations for the parallel MR imaging problem and the correct reference appears to be:
ESPIRiT — An Eigenvalue Approach to Autocalibrating Parallel MRI: Where SENSE meets GRAPPA, by Uecker, et al., 2014.
This is the reference from the fastMRI paper by Zbontar et al.

3.
In the case of CT imaging, 3D cone-beam CT or helical CT are the common practical data acquisition techniques and 2-D simulations are not particularly convincing.
Please refer to the following paper for a reasonable 3-D simulation as well as results on real sinogram data:
Kim, Donghwan, Sathish Ramani, and Jeffrey A. Fessler. "Combining ordered subsets and momentum for accelerated X-ray CT image reconstruction." IEEE transactions on medical imaging 34, no. 1 (2014): 167-178.
It may be acceptable that the authors do not simulate practically important effects such as beam hardening, but a 2-D simulation and an FBP baseline can be misleading.
It is also common practice in CT reconstruction to use an anthropomorphic digital phantom during simulation in order to uncover reconstruction artifacts that might get hidden
when using a reconstructed CT image, itself full of noise and other artifacts.

4.
Some typos exist in the paper, e.g. "fidility" at the bottom of page 5.

**Questions:**

Please note the inherent question underlying weakness 1.

---

> ### Author Response · Authors · 2023-11-22
> **Official Response of Submission7428**
>
> We thank the Reviewer for their constructive comments. Below are our point-by-point responses to Weaknesses and Questions:
>
> **Weakness 1:** We appreciate the Reviewer's observation regarding introducing two additional parameters, ζ and ρ, in our algorithm (MGDM sampling) compared to DDNM. In response to this concern, we have conducted additional experiments to investigate the impact of these parameters on the algorithm's performance. Our findings indicate the following: (a) Once the parameters ζ and ρ are appropriately tuned within specific ranges, the performance of our MGDM becomes stable, making the parameter tuning process easier. (b) Moreover, we have observed that the parameters once tuned on a small number of specific datasets, exhibit the ability to generalize well to unseen datasets. This suggests that it is not always necessary to perform parameter tuning for each subject, as the tuned values can provide satisfactory performance across different datasets. It's worth noting that hyperparameter tuning is a universal challenge in machine learning and not unique to our method. Our revised manuscript provides detailed insights into our parameter selection process, allowing readers to replicate it in their experiments. Moreover, our ablation study of Table 3 showed that even without the two additional steps involving ζ and ρ (i.e., Ours_{no_ir}), our new proximal projection step alone can already marginally outperform DDNM.
>
> **Weakness 2:** We appreciate the Reviewer's keen attention to detail. We acknowledge the error in our citation of the 'ESPIRIT' paper and apologize for this oversight. We have promptly replaced it with the correct reference, the paper by Zbontar et al. Furthermore, we have thoroughly reviewed all other references to ensure their accuracy.
>
> **Weakness 3:** We appreciate the Reviewer's feedback regarding the limitations of our 2D digital simulation compared to practical 3D cone-beam or helical CT data acquisitions. It's important to clarify that our choice of 2D simulation using parallel-beam geometry aligns with the methodology adopted by several existing methods, including diffusion models such as MCG [1], SIN-4c-PRN [2], and ScoreMed [3]. This allows for a direct comparison of algorithm performances.
>
> We acknowledge the limitations of this simplified 2D simulation and the absence of practical effects like beam hardening. To address these concerns more comprehensively, we have referenced the paper 'Combining ordered subsets and momentum for accelerated X-ray CT image reconstruction' for further discussion, as suggested by the Reviewer. In light of the short timeframe for revision, we understand that a 3D realistic CT phantom simulation is a valuable avenue for future work, and we appreciate the Reviewer's input in this regard.
>
> Quoted from our limitation part: “It should be noted that our CT simulation adheres to the 2D parallel beam geometry assumption, aligning with the baseline models used in other studies for direct comparison. This differs from the more complex and realistic 3D cone-beam CT or helical CT simulations (Kim et al., 2014).”
>
> **Weakness 4:** Thank you for pointing out the typographical errors in our manuscript. We have carefully reviewed the document and corrected the typo "fidility" to "fidelity" at the bottom of page 5, along with a thorough check for any additional typos throughout the text.
>
>
> [1] Hyungjin Chung, Byeongsu Sim, Dohoon Ryu, and Jong Chul Ye. Improving diffusion models for inverse problems using manifold constraints. Advances in Neural Information Processing Systems, 35:25683–25696, 2022b.
>
> [2] Haoyu Wei, Florian Schiffers, Tobias Wu ̈rfl, Daming Shen, Daniel Kim, Aggelos K Katsaggelos, and Oliver Cossairt. 2-step sparse-view ct reconstruction with a domain-specific perceptual net- work. arXiv preprint arXiv:2012.04743, 2020.
>
> [3] Yang Song, Liyue Shen, Lei Xing, and Stefano Ermon. Solving inverse problems in medical imaging with score-based generative models. arXiv preprint arXiv:2111.08005, 2021.

---

> > ### Comment · Reviewer_sAeb · 2023-11-23
> > **Thank you for your rebuttal !**
> >
> > I wish to thank the authors for their detailed response to my comments as well as those from other reviewers.
> > I especially appreciate the additional experiments regarding the tuning of additional hyper-parameters \zeta and \rho. Those additional results are not very surprising.
> > I would like to retain my original (positive) rating, although the concerns raised by one reviewer regarding the code and reproducibility of the results do appear to be valid.

---

### Meta-Review · Area_Chair_ePfH · 2023-12-06

**Metareview:**

This paper presents a method for inverse problem of CT and MRI reconstruction from incomplete and/or noisy scan data, which guides the diffusion process through a bi-level strategy. Although the authors show some improvement in the results, the reproducibility of the paper is questionable. The authors report wrong results at the original submission and updated after being questioned. The shared codes cannot be run right away by the reviewer TLju who had spent a lot of time and effort in reviewing the paper and response to the rebuttal.  Given these value points, I cannot recommend this paper.

**Justification For Why Not Higher Score:**

The paper has too many vague points.

**Justification For Why Not Lower Score:**

NA

---

### Decision · Program_Chairs · 2024-01-16

Reject